# Recurrent evolutionary switches of mitochondrial cytochrome *c* maturation systems in Archaeplastida

Huang Li [1], Soujanya Akella [1,2], Carina Engstler[3], Joy J. Omini[4], Moira Rodriguez [2], Toshihiro Obata [1,4], Chris Carrie [5], Heriberto Cerutti [1,2] & Jeffrey P. Mower [1,6] ✉

Mitochondrial cytochrome *c* maturation (CCM) requires heme attachment via distinct pathways termed systems I and III. The mosaic distribution of these systems in Archaeplastida raises questions about the genetic mechanisms and evolutionary forces promoting repeated evolution. Here, we show a recurrent shift from ancestral system I to the eukaryotic-specific holocytochrome *c* synthase (HCCS) of system III in 11 archaeplastid lineages. Archaeplastid HCCS is sufficient to rescue mutants of yeast system III and Arabidopsis system I. Algal HCCS mutants exhibit impaired growth and respiration, and altered biochemical and metabolic profiles, likely resulting from deficient CCM and reduced cytochrome *c*-dependent respiratory activity. Our findings demonstrate that archaeplastid HCCS homologs function as system III components in the absence of system I. These results elucidate the evolutionary trajectory and functional divergence of CCM pathways in Archaeplastida, providing insight into the causes, mechanisms, and consequences of repeated cooption of an entire biological pathway.

The repeatability of evolution and its potential adaptive significance has fascinated scientists as far back as Charles Darwin, and the relative role of genetic changes and contingent events in shaping the possible evolutionary outcomes is an ongoing focus of study[1–3]. This repeatability manifests as convergent evolution, which describes the independent evolution of a similar phenotype in different species[2,3]. Parallel evolution is sometimes distinguished from convergent evolution when a phenotype has repeatedly evolved via the same underlying mutation or process, although ambiguity arises in assessing the biological basis of the convergent phenotypes[4,5], potentially involving the same developmental alteration, pathway flux, shift in gene expression or even amino acid replacement. Regardless, both convergent and parallel evolution are usually framed in the context of adaptation, in which the recurrently evolved trait is assumed to repeatedly arise due

to an adaptive benefit in at least some contexts, such as the multiple origins of powered flight in animals[6], the repeated convergence of floral morphology among plants[7,8], the multiple independent shifts from C3 to C4 or CAM photosynthesis in plants[9,10], and the repeated convergence of various protein adaptations, such as hemoglobins that enable high-altitude respiration[11,12] and enzymes that produce plant secondary metabolites[13,14].

Yet, repeated evolution can also occur when the adaptive benefit is not apparent. One example relates to the maturation of *c*-type cytochromes, which are essential electron carriers in the electron transport chain (ETC) of mitochondria, chloroplasts, and prokaryotes. Cytochrome *c* maturation (CCM) requires the covalent attachment of a heme group to the apocytochrome, which occurs at conserved cysteines in a CXXCH motif to form the holocytochrome. CCM occurs

[1]Center for Plant Science Innovation, University of Nebraska-Lincoln, Lincoln, NE 68588, USA. [2]School of Biological Sciences, University of Nebraska-Lincoln, Lincoln, NE 68588, USA. [3]Department Biologie I–Botanik, Ludwig-Maximilians-Universität München, D-82152 Planegg-Martinsried, Germany. [4]Department of Biochemistry, University of Nebraska-Lincoln, Lincoln, NE 68588, USA. [5]School of Biological Sciences, University of Auckland, Auckland 1142, New Zealand. [6]Department of Agronomy and Horticulture, University of Nebraska-Lincoln, Lincoln, NE 68583, USA. ✉e-mail: jpmower@unl.edu

by at least three distinct pathways: system I and II originated in prokaryotes and persist in mitochondria (system I) or plastids (system II) of diverse eukaryotes, whereas system III is specific to mitochondria of some eukaryotic lineages, including animals, fungi, amoebozoans, and various algae[15,16]. The sharing of systems between bacteria and organelles indicates that mitochondria and plastids likely inherited system I and II from their alpha-proteobacterial and cyanobacterial progenitors. In contrast, the sporadic distribution of system III among eukaryotes implies a complex evolutionary history involving repeated independent shifts from system I to system III[15,16]. Moreover, increased mitogenomic sampling from diverse eukaryotes[17–22] has revealed that the loss of system I (and presumed shift to system III) is more prevalent than currently appreciated.

Although both CCM systems I and III catalyze heme attachment to mitochondrial $c$-type cytochromes, the two pathways differ in their components and complexity[15,16]. In *E.coli*, system I comprises one operon encoding eight Ccm proteins:[23] CcmA and CcmB form an ABC transporter; CcmC and CcmD deliver heme to CcmE, a heme chaperone; CcmF is the putative heme synthase that transfers heme from CcmE to the apocytochrome; and CcmG and CcmH participate in a thioreduction pathway that reduces the disulfide bond to allow heme attachment[24]. In plants, six system I homologs have been identified: CCMA, CCME and CCMH are usually encoded by the nuclear genome, while CCMB, CCMC and CCMF (CCMF is often split into two or three genes) are usually encoded by the mitogenome[25,26]. By contrast, the eukaryote-specific system III is comparatively very simple, relying on a holocytochrome $c$ synthase (HCCS) enzyme that binds heme and apocytochrome, catalyzes ligation, and releases the holocytochrome $c$[16,27]. In yeast, two HCCS homologs (CCHLp and CC$_1$HLp) operate specifically on apocytochromes $c$ and $c_1$, respectively, whereas in animals, a single HCCS can act on both $c$-type mitochondrial apocytochromes[28]. Because a defect in heme attachment has implications in mitochondrial pathology and disease, human HCCS has been extensively studied in terms of its structure, mitochondrial membrane localization and mechanism for generating the mature cytochrome $c$[29–32].

Nevertheless, our understanding of the evolutionary history of CCM systems among eukaryotes, especially the spread and functional divergence of the system III HCCS among distinct lineages, remains limited. The repeated and independent shifts of the mitochondrial CCM pathway from system I to III raises fundamental questions about repeated evolution via pathway cooption and the underlying genetic mechanisms and evolutionary forces that promote or deter pathway shifts in specific lineages. In particular, how did the same system III HCCS enzyme become so sporadically distributed among eukaryotes? The Archaeplastida (comprising land plants, green algae, red algae, and glaucophytes) provide an ideal test group for understanding the evolution of system I and III CCMs. Mitogenomic sequencing of lycophytes[33–35], hornworts[36–38], a liverwort[39], a fern[21], and various green and red algae[22,40,41] demonstrated that all mitochondrial system I genes were repeatedly lost during Archaeplastida evolution, although the number of independent examples has not been examined comprehensively. Importantly, in the transcriptome of the fern *Ophioglossum californicum*, no transcripts were detected for the missing mitochondrial *ccm* genes, arguing against their transfer to the nuclear genome, or for any nuclear *Ccm* genes, indicating that the entire system I pathway was lost[21]. In *Chlamydomonas reinhardtii*, which also lacks system I genes in its mitochondrial and nuclear genomes, two putative system III HCCS genes were detected in the nuclear genome[15,25,42], suggesting that the loss of system I was functionally coopted by a switch to system III (as in fungi and animals), although this inference awaits experimental confirmation.

In this work, to determine the number of shifts from system I to III in Archaeplastida, we conduct a comprehensive survey on the presence/absence of the mitochondrion-encoded members of the system I pathway and the nucleus-encoded components of systems I and III.

To test the inference of cooption of system I by system III in diverse Archaeplastida lineages, we characterize the function of HCCS homologs from the fern *Ceratopteris richardii*, an emerging non-flowering plant model in the study of plant evolution[43,44], and *Chlamydomonas reinhardtii*, a unicellular algal model amenable to a variety of genetic and biochemical analyses[45,46] and whose mitochondrial respiration-deficient mutants can be viably maintained by photosynthesis under light conditions[47,48]. These properties make *C. reinhardtii* an ideal model system for investigating mitochondrial respiration and carbon metabolism.

## Results

### Recurrent loss of CCM System I from diverse Archaeplastida

Mitochondrial genome sequencing has revealed the loss of *ccm* genes (*ccmB*, *ccmC*, *ccmF$_C$*, *ccmF$_N$*) encoding members of the system I CCM pathway from various lineages of land plants, green algae, and red algae[21,22,33–41]. To comprehensively evaluate the frequency and phylogenetic distribution of mitochondrial *ccm* gene losses across Archaeplastida, a survey of mitochondrial genomes from diverse land plants, green algae, red algae, and glaucophytes was performed, which identified at least 11 independent losses of the full suite of mitochondrial *ccm* genes (Fig. 1). These losses occurred in disjunct archaeplastid lineages, including (1) leptosporangiate ferns, (2) eusporangiate ferns from *Ophioglossum*, (3) all lycophytes, (4) all hornworts, (5) the liverwort *Treubia lacunosa*, (6) Chaetosphaeridiales, (7) the klebsormidiophyte *Entransia fimbriata*, (8) the common ancestor of *Chlorokybus* and *Mesostigma*, (9) all chlorophytes, (10) all red algae in Rhodophytina, and (11) all glaucophytes.

To assess the functional status of the nuclear *Ccm* genes (*CcmA*, *CcmE*, *CcmH*) encoding members of the system I pathway, and to evaluate whether the lost mitochondrial *ccm* genes were functionally transferred to the nucleus, we surveyed the presence of these system I genes in published nuclear genomes and newly assembled transcriptomes from diverse Archaeplastida (Fig. 1). For all species that lost their mitochondrion-encoded system I genes, we failed to detect any copies of the lost mitochondrial genes in their nuclear genome/transcriptome, arguing against functional transfer of these genes from the mitochondrion to the nucleus, and we failed to detect the suite of nucleus-encoded *Ccm* genes, which strongly implies complete loss of system I from these species. In contrast, all species that retained their mitochondrial *ccm* genes also retained their nuclear *Ccm* genes, indicating a presumably intact system I. Our blast-based survey results are likely to be robust because we used system I homologs from multiple Archaeplastida sequences as queries, our findings for the mitochondrial encoded proteins were corroborated by available mitogenome annotations, and the presence/absence of system I genes was concordant between the mitochondrial and nuclear genomes of each species, even for those with fast-evolving mitogenomes such as *C. reinhardtii*[41] and *Galdieria sulphuraria*[40]. Overall, our survey expands upon previously reported mitochondrial *ccm* losses by documenting at least 11 independent losses of the entire system I CCM pathway across Archaeplastida.

### Repeated shifts to CCM system III among Archaeplastida that lost system I

As the maturation of mitochondrial cytochrome $c$ and $c_1$ holoforms is likely an essential process, an obvious question is raised: how does CCM proceed in Archaeplastida lineages that lack system I? *C. reinhardtii* possesses two genes (*HCS1*, *HCS2*) encoding homologs to human and fungal HCCS, suggesting that this species has switched to the system III CCM pathway. Using the *C. reinhardtii* genes as queries, we identified HCCS homologs in the nuclear genomes of *Selaginella moellendorffii* and *Chondrus crispus*, which also lack evidence of system I. Using these diverse HCCS sequences, we surveyed the remaining archaeplastid nuclear genomes and assembled

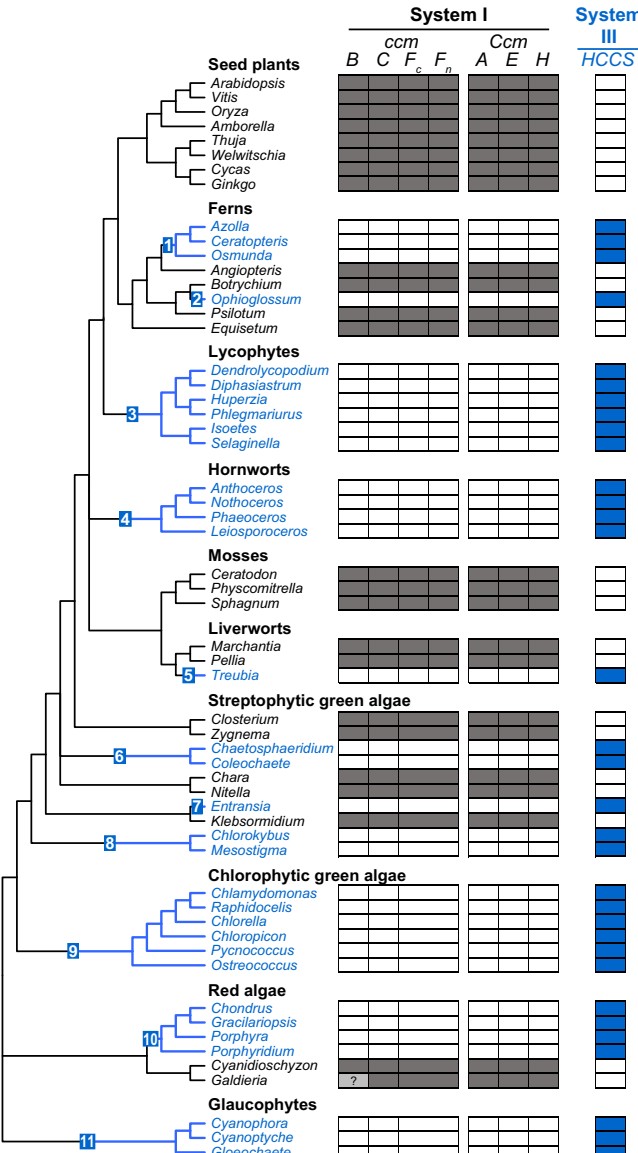

**Fig. 1 | Distribution of system I and system III genes for mitochondrial CCM in sequenced Archaeplastida species.** The boxes indicate the presence (black or blue), absence (white), or uncertain status (gray with question mark) of system I and III CCM genes in each species. Species in blue lack system I but possess system III, and the inferred shifts from system I to III were counted and mapped onto the phylogenetic tree, whose relationships are based on recent studies[93,94]. The status of *ccmB* is uncertain in *Galdieria*[40].

transcriptomes for the presence of HCCS homologs. A homolog to HCCS was identified in all species that have lost system I, but no HCCS was detected in species that retained system I (Fig. 1). This dichotomous pattern strongly suggests that the recurrent loss of system I in diverse Archaeplastida was enabled by the presence of system III HCCS.

### Plant and green algal HCCS localize to mitochondria and rescue yeast system III mutants

Although homologs to a system III HCCS were detected in all Archaeplastida lineages that have lost system I genes, no study has experimentally confirmed that any of these archaeplastid homologs are functional system III HCCS enzymes capable of maturation of mitochondrial cytochromes *c*. To assess whether *Ceratopteris* HCCS (CriHCCS) is targeted to the mitochondrion, a vector carrying the CriHCCS coding sequence fused with GFP was transiently

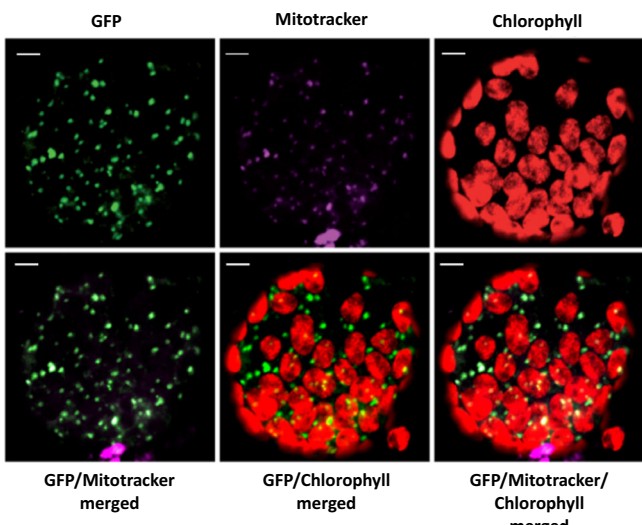

**Fig. 2 | Subcellular localization of *Ceratopteris* HCCS (CriHCCS) in tobacco leaf protoplasts.** The 35S:*CriHCCS*-GFP construct was transformed into tobacco leaves. GFP, MitoTracker signal and chlorophyll autofluorescence in the transformed tobacco cells were separately observed by confocal laser scanning microscopy and then resulting images were merged. The experiment was repeated three times with similar results. Scale bars, 5 μm.

agroinfiltrated into tobacco leaves from which protoplasts were isolated. Confocal laser-scanning microscopy revealed clear mitochondrial localization of the GFP fluorescence, as confirmed by counterstaining with MitoTracker (Fig. 2). The two *Chlamydomonas* homologs (CreHCS1 and CreHCS2) were also predicted to be localized to mitochondria (Supplementary Table 1).

In human and yeast CCM via system III, HCCS is responsible for heme attachment to *c*-type apocytochromes. *S. cerevisiae* carries two versions of this enzyme: CCHLp (encoded by *CYC3*) and $CC_1HLp$ (encoded by *CYT2*) attach heme to mitochondrial cytochromes *c* and $c_1$, respectively. Yeast cells with a deleted *CYC3* are completely, or almost completely, deficient in cytochrome *c* and, thus, do not grow on complete medium containing a nonfermentable carbon source such as glycerol[28]. This Δ*cyc3* strain, when transformed with *CriHCCS* constructed in a vector under the control of the constitutive promoter PGK1, was able to grow on glycerol medium after a 5-day incubation, suggesting that CriHCCS can functionally complement this respiratory deficient mutant and restore the capability of aerobic respiration (Fig. 3a). We also found that CriHCCS could complement the Δ*cyt2* yeast mutant deficient in holocytochrome $c_1$ synthase. In a parallel analysis of *C. reinhardtii* HCCS homologs, CreHCS1 was able to rescue the corresponding Δ*cyc3* strain, but not Δ*cyt2*, in the yeast complementation assay, while CreHCS2 could rescue Δ*cyt2* but not Δ*cyc3* (Fig. 3b).

### CRISPR-generated *Chlamydomonas* System III mutants exhibit growth defects

Wild-type *Chlamydomonas* can grow heterotrophically in the dark when the growth medium is supplemented with carbon sources, such as acetate, or photoautotrophically in the light using $CO_2$ as the carbon source[45,49]. Taking advantage of this characteristic, a number of respiratory-deficient strains for different ETC components, including complex I (NADH dehydrogenase), complex II (succinate dehydrogenase), complex III (ubiquinol: cytochrome *c* oxidoreductase), and complex IV (cytochrome *c* oxidase), have been isolated by mutagenesis in combination with phenotypic observation of their null or slow growth in heterotrophic conditions[48,50,51]. However, *Chlamydomonas* mutants with altered cytochrome *c* function are limited to

mitochondrial genes, such as *cob* (encoding apocytochrome *b* of complex III)[50,51] and *cox1* (encoding the subunit 1 of complex IV)[52].

Here, to further evaluate the function of *Chlamydomonas HCS1* and *HCS2*, two nucleus-encoded genes for system III *c*-type cytochrome

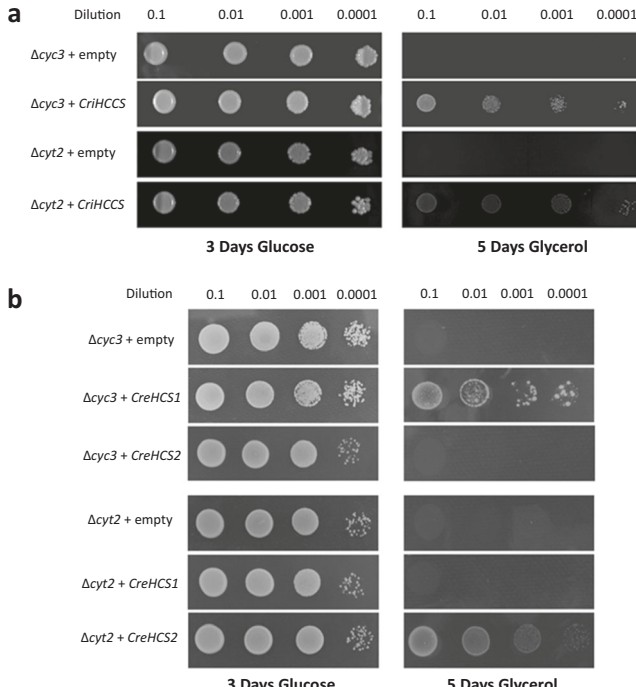

**Fig. 3 | Yeast complementation with *Ceratopteris* HCCS (CriHCCS) and *Chlamydomonas* HCS (CreHCS1 and CreHCS2) homologs.** *S. cerevisiae* strain Y00367 (Δ*cyc3*; CCHLp) and Y04936 (Δ*cyt2*; CC₁HLp) were transformed with a plasmid containing **a** *Ceratopteris* CriHCCS coding sequence or **b** *Chlamydomonas CreHCS1* or *CreHCS2* coding sequence, each under control of the PGK1 promoter. An empty vector was used as a negative transformation control. Equal amounts of serial dilutions of cells from exponentially grown cultures were spotted onto solid synthetic defined (SD) plates supplemented with 2% glucose or 3% glycerol and incubated at 25 °C. Each spot assay was repeated three times with similar results.

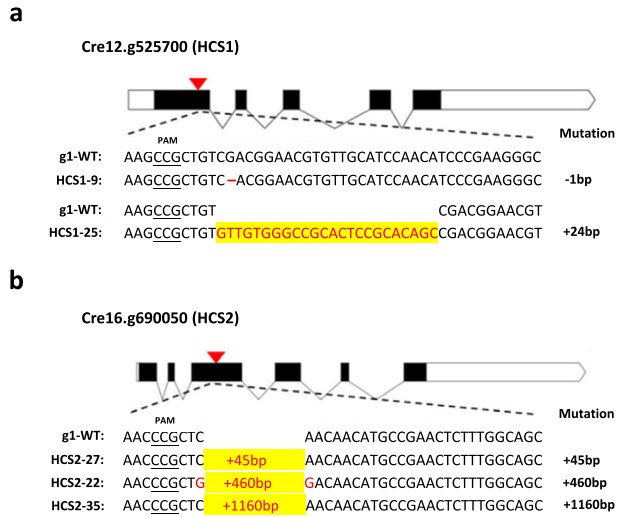

maturation, we obtained several independent *CreHCS1* and *CreHCS2* mutants by CRISPR-Cas9 mediated gene editing (Fig. 4). PCR amplification and sequencing of the HCS loci confirmed the occurrence of targeted mutations at the expected *CreHCS1* and *CreHCS2* CRISPR-Cas9 RNP cut sites. Among the sequenced mutants, HCS1-9 carries one nucleotide deletion in exon 1, resulting in a frameshift that leads to a premature stop codon and, consequently, a functional knock-out of *CreHCS1*; HCS1-25 has a 24 bp insertion (Fig. 4a). In HCS2 editing, two mutants (HCS2-22, HCS2-35) obtained from dark screening have large insertions (460 bp and 1160 bp, respectively), which likely disrupt the function of HCS2. Interestingly, HCS2-27, a colony identified by PCR screening before the implementation of dark screening, has an in-frame mutation (45 bp insertion) (Fig. 4b).

We next compared the growth of the WT strain (g1) and the HCS1 and HCS2 mutants cultivated in acetate-containing culture medium (TAP) under continuous light or dark conditions (Fig. 4c). During the first three days of light exposure, the knock-out mutants HCS1-9 and HCS2-22 exhibited a slower growth rate compared to that of WT. In dark conditions, HCS1-9 and HCS2-22 failed to grow, while WT showed continuous growth, although its rate began to decline after day three. HCS2-27 mutant behaved comparably with WT under light but grew slower in the dark, indicating a partial functional impairment in its heterotrophic growth caused by the in-frame mutation. Its dark-growth defect was confirmed in a spot assay (Supplementary Fig. 1a).

## *Chlamydomonas* HCS mutants are defective in cytochrome *c*-dependent respiratory chain with altered TCA intermediate accumulation

TTC (2,3,5-triphenyltetrazolium chloride) is a colorless molecule that can be converted to formazan (a red compound) by the mitochondrial ETC with a functional cytochrome pathway[53,54]. When treated with TTC, *Chlamydomonas* WT g1 and the HCS2-27 cells turned dark brown (a mixture of green and red color), whereas the knock-outs HCS1-9 and HCS2-22 remained green (Supplementary Fig. 1b), indicating a disruption of mitochondrial ETC activity in the knock-out mutants due to the functional impairment of cytochrome pathway components (complex III and/or IV).

We also measured oxygen consumption rates of the cells grown in heterotrophic conditions (dark + acetate). Like land plants,

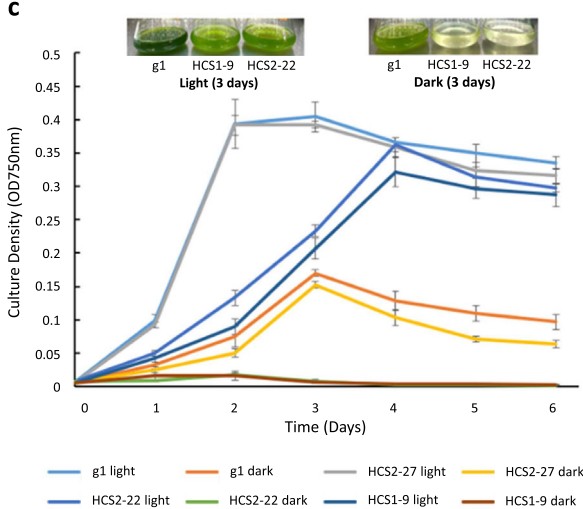

**Fig. 4 | Description and growth measurement of *Chlamydomonas* HCS mutants generated by CRISPR-Cas9.** **a** Schematic of the *CreHCS1* target site (red arrow) and the mutated region (shown in red) in mutant HCS1-9 (1 bp deletion) and HCS1-25 (24 bp insertion). **b** Schematic of the *CreHCS2* target site (red arrow) and the insertions (the number of inserted nucleotides was labeled in red) of mutant HCS2-27, HCS2-22, and HCS2-35. **c** Growth curve of g1 and HCS mutant strains cultivated in TAP medium under light and dark conditions. Values shown are the average of three biological replicates ± SD. Pictures are representative of strains on day 3 from three experimental replicates. Source data are provided as a Source Data file.

*Chlamydomonas* has two respiratory pathways: the classical cytochrome oxidative phosphorylation pathway and an alternative pathway that branches from the ubiquinol pool and directly reduces $O_2$ to $H_2O$ without passing through the cytochrome pathway. The classical cytochrome pathway is sensitive to cyanide (NaCN), whereas the alternative pathway is resistant to cyanide but sensitive to salicylhydroxamic acid (SHAM) (Fig. 5). The HCS1 and HCS2 knock-out mutants (HCS1-9, HCS2-22) exhibited nearly 70% reduction in total dark respiration rate (Table 1), in agreement with our observation that both mutants exhibited severe growth defects in the dark (Fig. 4c). The addition of cyanide alone reduced the respiration of WT cells to around 30%, but cyanide had much less effect on the respiration of the knock-out strains. In contrast, the addition of SHAM abolished the oxygen consumption in the knock-out mutants (Table 1). The sensitivity to SHAM but insensitivity to cyanide in the knock-out mutants indicate that their reduced respiration rate was due to the dysfunction of the cytochrome pathway (cyanide-sensitive), whereas the

alternative oxidase pathway (SHAM-sensitive) is still functional. The HCS2-27 mutant had a dark respiration rate in between the WT and knock-out strains and showed intermediate effects in response to the

**Table 1 | In vivo measurement of dark respiration and the impact of SHAM and NaCN on the respiratory pathways of *Chlamydomonas* HCS mutants**

| Strain | Dark respiration | + NaCN | + SHAM | + SHAM + NaCN |
|---|---|---|---|---|
| WT (g1) | 14.9 ± 0.9[a] | 5.3 ± 0.6 | 12.1 ± 0.8 | 2.0 ± 0.3 |
| HCS2-27 | 11.2 ± 1.2 | 6.0 ± 0.3 | 9.7 ± 0.6 | 1.4 ± 0.2 |
| HCS2-22 | 4.5 ± 0.5 | 4.0 ± 0.2 | 0 | nm |
| HCS1-9 | 5.2 ± 0.4 | 2.8 ± 0.2 | 0 | nm |

*NaCN* sodium cyanide, *SHAM* salicylhydroxamic acid; nm, not measured. Source data are provided as a Source Data file.
[a]nmol $O_2$ min$^{-1}$ $10^{-7}$ cells.

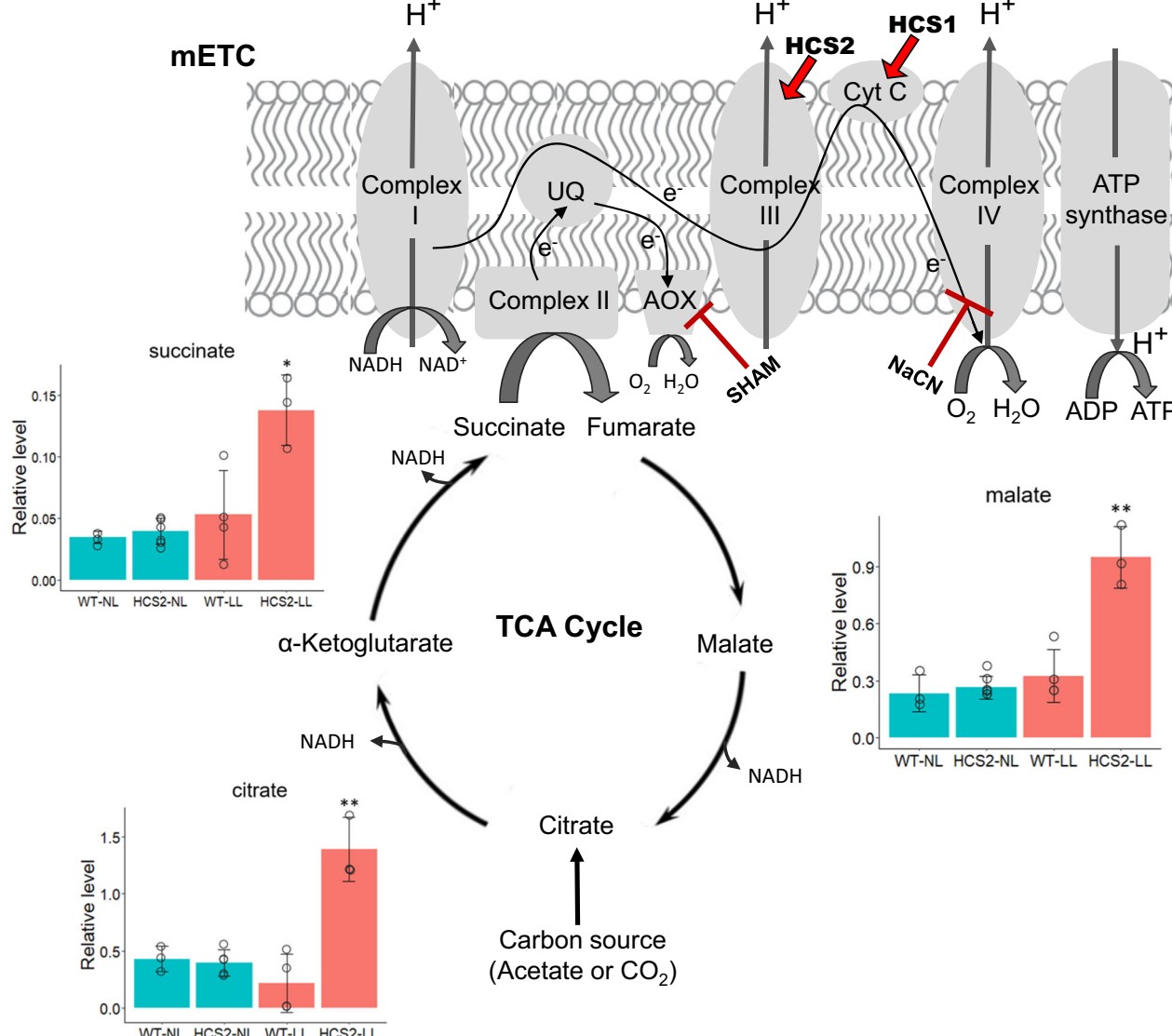

**Fig. 5 | A diagram showing the mitochondrial electron transport chain (mETC), the effect of HCS genes and inhibitors on ETC components, and the associated tricarboxylic acid (TCA) metabolic pathway in *Chlamydomonas*.** Red arrows indicate the involvement of HCS1 and HCS2 in the maturation of cytochrome *c* (CytC) and cytochrome $c_1$ (subunit of complex III) within the cytochrome oxidase pathway, which can be inhibited by cyanide (NaCN), independent from the SHAM-sensitive alternative oxidase pathway (AOX). Diagram of the TCA cycle with bar

graphs showing levels of metabolites in the wild type (WT) and HCS2-27 mutant (HCS2) growing in TAP medium under normal light (NL; blue bars) or low light (LL; orange bars) conditions. Data are presented as mean values +/− SD ($n$ = 3, 6, 4, 3. $n$ refers to biological replicates of each strain). Asterisks indicate significant differences between means evaluated by one-way ANOVA followed by Tukey's multiple comparisons test (*$p < 0.05$; **$p < 0.001$). Source data are provided as a Source Data file.

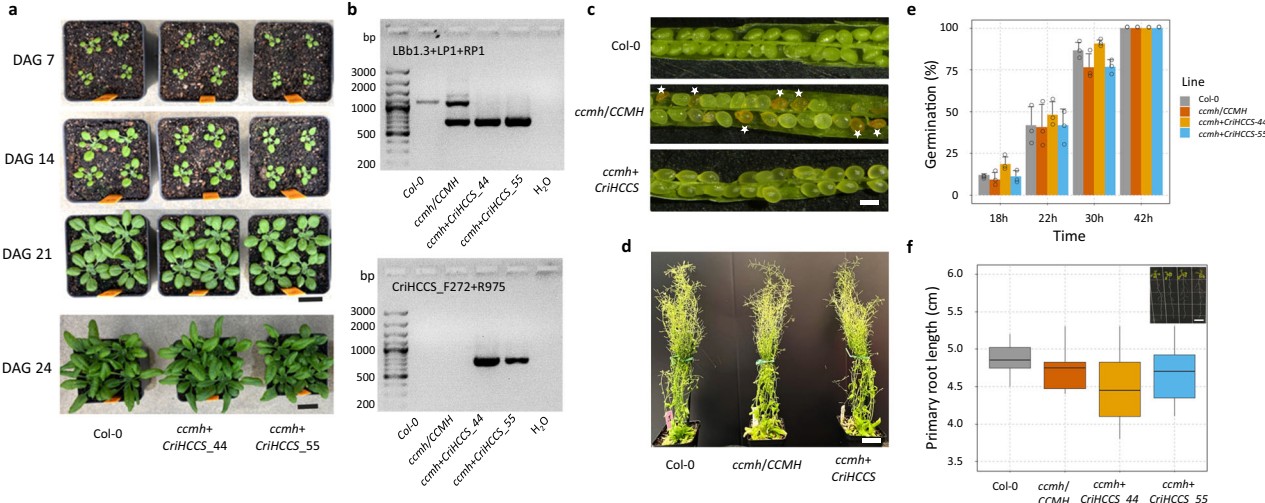

**Fig. 6 | System III HCCS is sufficient to complement Arabidopsis system I mutant. a** Transgenic expression of *CriHCCS* rescued the seed lethality of the Arabidopsis CCMH mutant, resembling the developmental growth of the wild type Col-0. DAG, Days after Germination. Scale bar = 2 cm. **b** Genotyping analysis of *ccmh* complementation line. Transformants were PCR genotyped for *ccmh* T-DNA homozygosity using three primers (LBb1.3, T-DNA left border sequence; LP1 and RP1, Left and right primer flanking the T-DNA insertion site in the *CCMH* gene) and for the presence of the inserted *pUBQ10:CriHCCS* complementation construct using primers located within the *CriHCCS* coding sequence (F272 and R975). The analysis was repeated three times with same results. **c** Abnormal phenotype of homozygous seeds (denoted by asterisks) in the silique of self-fertilized hetero-zygous *ccmh/CCMH* plants was rescued in the *ccmh+CriHCCS* plant. Scale bar = 500 μm. **d** Photo of representative 60-day-old plants in long-day growth conditions. Scale bar = 2 cm. **e** Germination of seeds from Col-0, heterozygous *ccmh/CCMH* plants, and *ccmh+CriHCCS* plants at different times after transfer to germination conditions. The data are presented as means ± sd and the circles represent the number of values analyzed (*n* = 3 independent experiment). **f** Primary root length of these lines mentioned above. The boxplots show the minimum, first quartile, median, third quartile, and maximum primary root length of each line (*n* = 12). One-way ANOVA followed by Tukey's multiple comparisons test was conducted to examine the variation. A representative picture of plants (corresponding to the lines in the boxplot from left to right) grown vertically on the MS plate was taken on day 14 after seed sowing. Scale bar = 1 cm. Source data are provided as a Source Data file.

inhibitors (Table 1), indicating a less severe effect on the cytochrome *c* oxidase pathway, in agreement with the results of growth measurement and TTC staining.

To further capture the impact of HCS mutations on the mitochondrial ETC, we performed metabolite profiling of the WT strain and the mutant strain HCS2-27 grown at normal and low light conditions. As suggested in our early experiments, HCS2-27 partially retained its mitochondrial function to support its heterotrophic growth and thus can be cultivated in parallel with the background WT strain for comparative metabolomics analysis. Using GC-MS analysis, we quantified a total of twenty-three primary metabolites, including sugars, amino acids, and the tricarboxylic acid (TCA) cycle intermediates (Supplementary Data 1). Under normal light conditions, no statistically significant differences in metabolite levels were detected between WT and HCS2-27, whereas in low light, the HCS2-27 strain accumulates several TCA cycle intermediates, namely succinate, malate, and citrate to higher levels than WT (Fig. 5). These results correlate with the observed dark-specific growth defects (Fig. 4c; Supplementary Fig. 1a) and the fact that mitochondrial respiratory defects in *Chlamydomonas* can be compensated for by photosynthesis under optimal illumination[48,55]. The data also indicate that the reduced activity of HCS2 alters mitochondrial metabolism, as evidenced by the fact that only the metabolites closely related to the TCA cycle were affected (Supplementary Fig. 2). More broadly, these CRISPR-generated mutants with novel defects in the mitochondrial cytochrome pathway will be valuable resources for the *Chlamydomonas* community, particularly in advancing research within the field of algal organelle biology.

### System III HCCS is sufficient to complement an *Arabidopsis* system I mutant
The yeast complementation results demonstrate that archaeplastid HCCS homologs can function in other eukaryotes using system III (Fig. 3). Characterization of *Chlamydomonas* HCS mutants further

confirmed their primary roles in the system III pathway (Figs. 4, 5). In plants, we hypothesize that CCM by system I is being repeatedly replaced by system III HCCS. To investigate whether a lethal system I CCM mutant can be functionally complemented by system III HCCS in plants, we used an Arabidopsis T-DNA insertion line, *ccmh* (SALK_046872) for transgenic study (Fig. 6). CCMH plays a central role as a mitochondrial thiol-disulfide oxidoreductase in cytochrome c maturation. Due to the essential function of CCMH, only heterozygosis of the T-DNA insert was found in the progeny of *ccmh* line, consistent with a previous report that *ccmh* mutation is lethal in its homozygote state[56]. Thus, we transformed heterozygous *ccmh/CCMH* plants with a system III HCCS construct (*pUBQ10::CriHCCS*). Progeny from the transformed *ccmh/CCMH* plants were selected and genotyped for the presence of the HCCS construct and the homozygous *ccmh* mutation (Fig. 6b). Seeds harvested from independent T3 lines (*ccmh+CriHCCS*) were used for further evaluation.

The growth rate of complemented plants resembled that of the WT under greenhouse conditions (Fig. 6a). Moreover, the abnormal seeds in the immature siliques of a heterozygous *ccmh/CCMH* plant were not observed in the complemented lines (Fig. 6c), suggesting the system III HCCS successfully rescued the embryo-lethality of *ccmh* homozygosity. The homozygous system I mutant line, rescued with the system III HCCS, produced seeds and was phenotypically indistinguishable from WT (Fig. 6d). Plate-based phenotyping demonstrated that all lines exhibited similar seed germination rates, measured as the percentage of seeds with visible root protrusion, after 18–42 h of transfer to germination conditions (Fig. 6e). Similarly, the root growth of *ccmh+CriHCCS* seeds exhibited no differences relative to the WT (Fig. 6f). Taken together, our complementation study revealed that exogenous expression of a system III HCCS gene is sufficient to rescue the seed lethality of Arabidopsis system I CCMH mutant and complement its essential function for producing viable seeds and enabling normal plant growth.

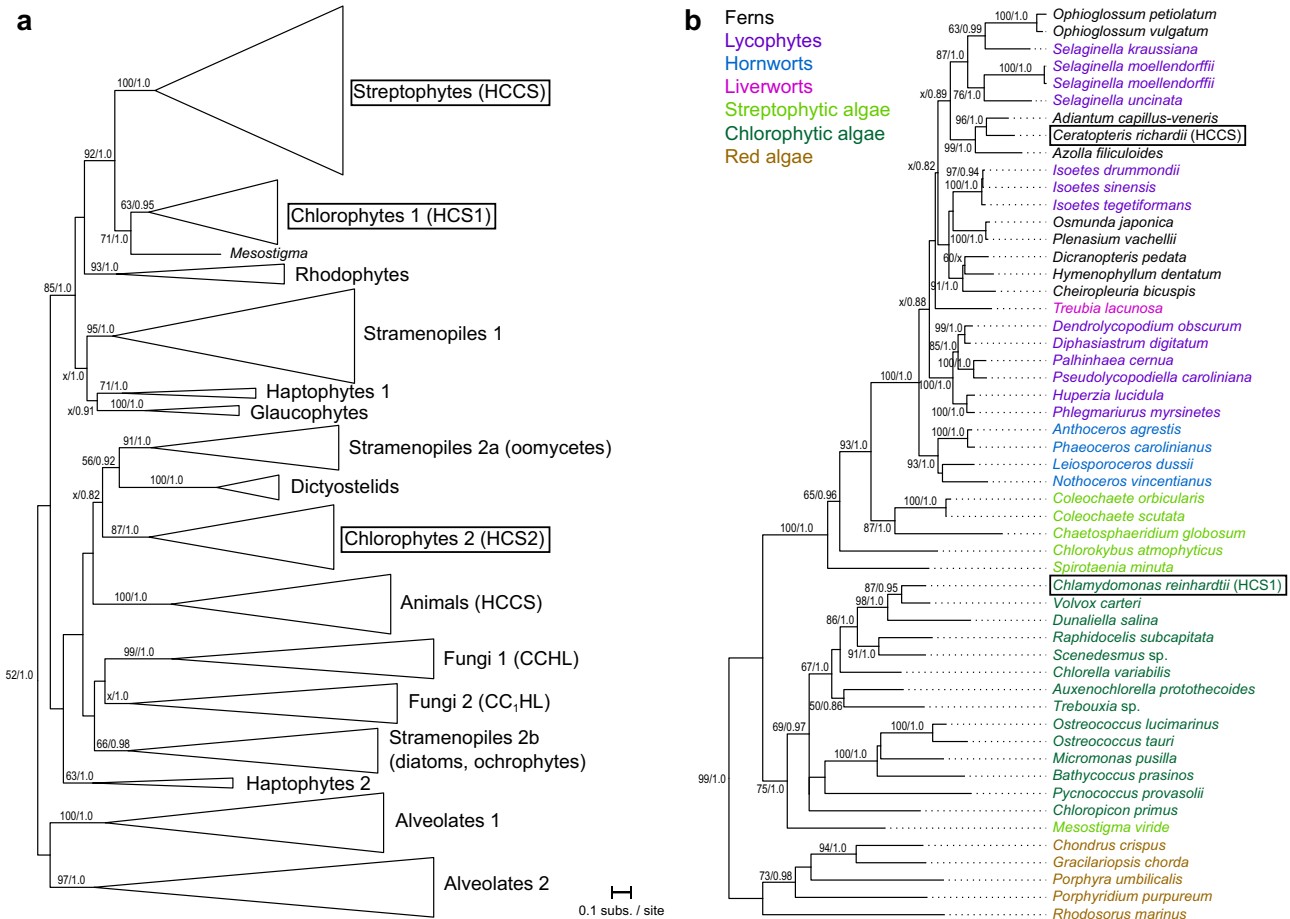

**Fig. 7 | Phylogenetic analysis of HCCS homologs. a** Phylogeny of eukaryote HCCS homologs with branches compressed into major groups. The tree was rooted using midpoint rooting. The fully labeled ML tree is shown in Supplementary Fig. 3, and the Bayesian tree is shown in Supplementary Fig. 4. **b** Phylogeny of HCS homologs in land plants, green and red algae. The tree was rooted on red algae, based on the results in Fig. 7a. For both trees, bootstrap values < 50% and posterior probabilities <0.8 are not shown. Source data are provided as a Source Data file.

## Phylogenetic analysis of eukaryotic HCCS homologs

The CCM survey demonstrated that all HCCS homologs detected among Archaeplastida are related in sequence (Fig. 1), and the functional assays demonstrated that fern HCCS and *Chlamydomonas* HCS1 and HCS2 are functional components of a system III pathway, capable of maturation of *c*-type apocytochromes (Figs. 2–6, Table 1). However, it is unclear how so many Archaeplastida lineages independently switched from system I to III, raising questions about the evolutionary mechanisms promoting these multiple pathway shifts in Archaeplastida and about the evolution of HCCS proteins capable of either bifunctional maturation of both cytochrome *c* and *c₁* or monofunctional maturation of only one *c*-type cytochrome.

Thus, we performed a phylogenetic analysis of Archaeplastida HCCS homologs in the broader context of HCCS homolog diversity across eukaryotes (Fig. 7a). Notably, each of the major eukaryotic lineages (as labeled in the figure) is monophyletic, except for stramenopiles group 2 for which oomycetes (stramenopiles 2a) did not cluster with diatoms and ochrophytes (stramenopiles 2b). The HCCS proteins from streptophytes (including the bifunctional fern HCCS) strongly clustered (92%/1.0 ML bootstrap/Bayesian posterior probability) with chlorophytes group 1 (including the monofunctional HCS1 from *Chlamydomonas*). These proteins also associate with a larger clade that includes the presumably bifunctional proteins from red algae and glaucophytes and one HCCS copy from stramenopiles and haptophytes (85%/1.0). In contrast, the chlorophyte group 2 proteins, including the

monofunctional *Chlamydomonas* HCS2, are distant paralogs of these other Archaeplastida proteins. Chlorophytes group 2 instead associates with other eukaryotic groups including dictyostelids and stramenopiles group 2a. Overall, the bifunctional Archaeplastida HCCS enzymes are closely related to chlorophyte HCS1 (specific for apocytochrome *c*), whereas chlorophyte HCS2 (specific for cytochrome *c₁*) is a distant paralog.

Within Archaeplastida, the switch from system I to III has occurred at least 11 times (Fig. 1), prompting inquiries into the evolutionary relationships of HCCS homologs among Archaeplastida lineages that have switched to system III. At a broad level, HCCS relationships among Archaeplastida (Fig. 7b) are consistent with organismal relationships: chlorophytes (69%/0.97) and streptophytes (100%/1.0) each form a monophyletic group, except that the streptophyte *Mesostigma* grouped anomalously with chlorophytes (75%/1.0). Within streptophytes, the green algae form the expected grade, with Chlorokybales diverging earlier than the Coleochaetales relative to land plants (93%/1.0). Within land plants, however, ferns and lycophytes do not form expected monophyletic groups. Although branch support for many of these nonmonophyletic groupings was limited, *Ophioglossum* (a fern) is nested within a clade for *Selaginella* (a lycophyte) with strong branch support (87%/1.0). AU topology tests, forcing either fern monophyly, lycophyte monophyly, or fern and lycophyte monophyly, were rejected ($P < 0.05$), providing significant support against vertical inheritance and suggesting a history of horizontal transfer for HCCS among ferns and lycophytes.

## Discussion

Here, in a survey for system I and III CCM genes in genomes and transcriptomes of diverse Archaeplastida, we documented at least 11 independent shifts from system I to III in distinct lineages (Fig. 1). This inference was based on the losses of nucleus- and mitochondrion-encoded system I genes from five land plant lineages (leptosporangiate ferns, the eusporangiate fern *Ophioglossum*, all lycophytes, all hornworts, and the liverwort *Treubia*), three streptophytic green algal lineages (Coleochaetales, the klebsormidiophyte *Entransia*, and the basal lineage *Chlorokybus* + *Mesostigma*), all chlorophytes, all red algae in Rhodophytina, and all glaucophytes. The switch from system I to III was inferred by the perfectly parallel presence of HCCS homologs in all archaeplastid lineages that lack system I, but not in any lineages that retained system I (Fig. 1), and by functional assays, which showed that plant HCCS homologs are targeted to mitochondria (Fig. 2; Supplementary Table 1) and are sufficient to functionally complement mutants of yeast system III (Fig. 3) and Arabidopsis system I (Fig. 6). Moreover, *Chlamydomonas* HCCS mutants exhibit reduced growth (Fig. 4; Supplementary Fig. 1a), impaired mitochondrial activity (Table 1; Supplementary Fig. 1b), and altered mitochondrion-related metabolic flux (Fig. 5), demonstrating that these identified HCCS homologs are indeed necessary for growth and mitochondrial function, particularly under heterotrophic conditions. These recurrent CCM pathway shifts in Archaeplastida greatly expand our understanding of the evolution of CCM pathways across eukaryotes, for which variation of the usage of systems I and III has also been documented, albeit to a lesser extent, in Opisthokonta[20], Alveolata[57,58], Cryptista[59,60], and Discoba[61,62].

Interestingly, within Archaeplastida, most lineages have a single HCCS that presumably catalyzes the maturation of apocytochromes *c* and $c_1$. This inference was verified for fern HCCS, which could rescue yeast Δ*cyc3* and Δ*cyt2* deletion mutants (Fig. 3a). In chlorophytes, however, there are two divergent HCCS paralogs, and functional analyses of *Chlamydomonas* HCS1 and HCS2 showed that each homolog can only rescue one of two yeast system III mutants (Fig. 3b), indicating that each *Chlamydomonas* enzyme is specific for the maturation of either apocytochrome *c* or $c_1$, but not both. The generalized or specialized functions of HCCS in Archaeplastida are also echoed across eukaryotes, where some eukaryotes (including animals and dictyostelids) have a single HCCS that is presumed, and experimentally confirmed for human HCCS[28], to act on both apocytochromes, whereas many other eukaryotic lineages (including fungi, stramenopiles, haptophytes, and alveolates) have two distinct copies that are likely specific for the maturation of either apocytochrome *c* or $c_1$, as demonstrated for yeast CCHLp and $CC_1$HLp[63,64].

From a mechanistic standpoint, the repeated CCM pathway shifts from system I to III, and the variable usage of either generalized or specialized HCCS enzymes, raise baffling questions about the repeatability of evolution. To explain how nearly a dozen different plant and algal lineages, and even more eukaryotic lineages, have independently switched in parallel to a homologous HCCS enzyme, two competing hypotheses have been proposed:[15,25] systems I and III were both present in an early eukaryotic ancestor followed by differential loss in descendent lineages, or rampant horizontal gene transfer (HGT) spread HCCS in the disparate eukaryotic lineages that use system III. The explanation that both systems were ancestrally present and maintained over long evolutionary time periods, only for one or the other pathway to be lost in essentially all present-day lineages, is difficult to accept. To date, only a single eukaryotic lineage (the newly described Provora) has been documented to have both systems I and III[65,66], and this inference was based only on sequencing (rather than functional) data. If both pathways were compatible for much of eukaryotic evolution, then why do we not find more living lineages that have both systems today? And why would two redundant pathways be maintained for so long? The alternative possibility that the HCCS gene

has been repeatedly transferred horizontally among eukaryotes requires an extent of HGT that is likely to be contentious e.g.[67].

To tease these competing theories apart, we used phylogenetic analyses (Fig. 7). Importantly, the HCCS homologs form a monophyletic group for each major eukaryotic lineage (or two weakly split subclades in the case of stramenopiles group 2), indicating that HCCS was already present in the common ancestor of each of these ancient eukaryotic groups (Fig. 7a). Similarly, the focused analysis of HCCS and HCS1 homologs among green plants (Fig. 7b) revealed consistent patterns at the deepest levels, in which chlorophytes and rhodophytes are each monophyletic, and streptophytes are monophyletic except for *Mesostigma*, which groups instead as sister to chlorophytes. Within land plants, however, HCCS relationships were inconsistent with organismal relationships: lycophytes and ferns are not monophyletic, but instead, *Ophioglossum* is nested within *Selaginella* while *Isoetes* associates with some leptosporangiate ferns. Thus, our phylogenetic analyses suggest a possible compromise, in which systems I and III may have been present in the archaeplastid common ancestor, prompting early decisions to retain one or the other system, followed by more recent HGT that prompted more recent system I to III switches among land plants.

We also assessed the evolutionary relationships among the single HCCS and dual HCCS homologs found among eukaryotes (Fig. 7a). In Archaeplastida, chlorophytes have two functionally distinct paralogs (HCS1 and HCS2 in *Chlamydomonas*), whereas all other Archaeplastida species that use system III have a single bifunctional HCCS. Interestingly, chlorophyte HCS1 (which is specific for apocytochrome *c*) strongly associates with the bifunctional HCCS from other Archaeplastida and with one HCCS enzyme from stramenopiles and haptophytes (whose apocytochrome affinities are unknown). In contrast, chlorophyte HCS2 (which is specific for apocytochrome $c_1$) is a distant paralog of the Archaeplastida HCCS and HCS1 enzymes, instead more closely affiliating with the presumably bifunctional HCCS from slime molds and the second HCCS copy from stramenopiles group 2 A. This pattern suggests two possibilities. In one scenario, chlorophyte HCS1 evolved by specialization of the ancestral bifunctional HCCS, while chlorophyte HCS2 was acquired via HGT (although the exact donors can't be gleaned from the tree due to limited branch support). Alternatively, it is possible that the ancestral eukaryote already had two distinct HCCS isoforms, and the functional generalization of one gene allowed for the loss of the other in most archaeplastids, whereas chlorophytes retained both ancestral paralogs as monofunctional enzymes. Functional characterization of the HCCS paralogs from diverse eukaryotes is needed to distinguish these possibilities.

This recurrent shifting from system I to III in so many eukaryotic lineages raises fundamental questions about the evolutionary forces driving repeated pathway shifts. First, the adaptive significance of the CCM pathway shift to system III is not apparent, whereas nearly all other examples of repeated evolution (e.g., powered flight, $C_4$ and CAM photosynthesis, secondary metabolites) have an obvious adaptive benefit. From a functional standpoint, both system I and III catalyze the attachment of heme in the same orientation and position to the apocytochromes, resulting in identical holocytochrome end products, and our Arabidopsis HCCS lines indicate that a switch to system III has no obvious phenotype under optimal greenhouse conditions (Fig. 6). In this sense, if both systems are present in a lineage (either from vertical inheritance or horizontal acquisition), then the ultimate retention of system I or III may indeed be an arbitrary evolutionary coin flip with no functional consequences. On the other hand, system III is less complex than system I. Perhaps the simpler system III pathway is beneficial for many eukaryotes, while the increased complexity of system I may allow enhanced regulatory control over the mitochondrion in other eukaryotes. For example, increased mitochondrial complexity has been hypothesized as a regulatory control mechanism for the mitochondrial function of seed plants, for which mitochondria

play an essential role in particular developmental stages, including pollen development, germination and early seedling growth[68,69]. Others have speculated that the switch to system III in eukaryotes was facilitated by the reduced need for diverse substrate specificity, as mitochondria require heme attachment to only two *c*-type cytochromes, whereas prokaryotes require the diverse substrate specificity of system I to attach heme to many more *c*-type cytochromes[15].

Second, whereas the many examples of adaptive convergent evolution likely involve genetic or spatiotemporal regulatory changes to existing genes and pathways, the CCM shift from system I to III involves the swapping of two unrelated pathways. The most similar example to the CCM pathway evolution described here may be found in the recurrent replacement of translation elongation factor EF-1α by a related EF-Like protein, a switch that may not have any adaptive benefit and likely occurred either by HGT or differential loss of ancestrally coexisting factors[70,71]. Yet, even in that case, the two elongation factors are evolutionarily related, and the shift involves the cooption of a single enzyme rather than the cooption of the entire CCM system I pathway by an evolutionarily distinct system III pathway. Thus, our CCM work offers a unique view into the processes that can lead to recurrent evolutionary change at the pathway level and provides a tantalizing system to assess the underappreciated role of chance in recurrent evolution.

## Methods

### Survey of system I and system III genes for cytochrome *c* maturation

Sequences for system I genes were obtained from the annotated nuclear and mitochondrial genomes of *Arabidopsis thaliana* (land plant), *Cyanidioschyzon merolae* (red alga), and *Klebsormidium nitens* (green alga), while system III gene sequences were obtained from the nuclear genomes of *Chlamydomonas reinhardtii* (green alga), *Chondrus crispus* (red alga), and *Selaginella moellendorffii* (land plant) (Supplementary Table 2). Translated protein sequences of these system I and III genes were used in a tblastn v2.2.31 search (length > 100 bp; e-value < 1e$^{-20}$) to query their presence/absence in available nuclear genome annotations, available mitochondrial genomes, and newly assembled transcriptomes and draft mitogenomes from diverse Archaeplastida (Supplementary Data 2). To assemble draft mitochondrial genomes, Illumina data sets were assembled with a range of kmer values (61,71,81,91) and the careful option in SPAdes v3.13.0[72]. To assemble transcriptomes, RNA-seq data sets were assembled with default settings in Trinity v2.6.6[73].

### Subcellular localization of *Ceratopteris* HCCS and *Chlamydomonas* HCS

The *Agrobacterium tumefaciens* strain AGL1 was transformed with the full coding sequence of *CriHCCS* fused to the N-terminus of GFP under the control of the CamV 35 S promoter. Cloning was performed by Golden Gate cloning[74,75]. Transformed AGL1 were used to infiltrate 4–6 week old *Nicotiana benthamiana* leaves for transient expression of the gene construct. Briefly, the incubated AGL1 cell culture was resuspended in infiltration medium (10 mM MgCl$_2$, 10 mM MES pH 5.7, 150 μM acetosyringone) to an OD$_{600}$ = 1.0 and delivered to the underside of leaves using a blunt tipped syringe. Protein expression was monitored after 48 hours[76]. Protoplasts were prepared from leaves by cutting them into 5–10 mm squares in 10 ml of F-PIN (10 mM KNO$_3$, 3 mM CaCl$_2$, 1.5 mM MgSO$_4$, 1 mM KH$_2$PO$_4$, 20 mM ammonium succinate, 2 mM MES, Murashige-Skoog basal salts, adjusted to 550 mOsm with sucrose, pH 5.7) containing 1% cellulase Onozuka R-10, 0.3% macerozyme R-10, and 0.1% BSA. After shaking for 90 min at 40 rpm in the dark, protoplasts were released with shaking at 150 rpm for 1 min. The filtered suspension was overlaid with 2 ml F-PCN (same as F-PIN but adjusted to 550 mOsm with glucose) and centrifuged for 10 min at 70 × g with low acceleration and deceleration. Intact protoplasts were collected from the interphase and washed with W5 buffer (150 mM NaCl, 125 mM CaCl$_2$, 5 mM KCl, 2 mM MES, 550 mOsm, pH 5.7) by centrifuging for 10 min. Intact protoplasts were resuspended in 0.5 ml W5 buffer prior to microscopy[77]. Mitotracker (Life Technologies), a mitochondrial-specific dye, was added to the protoplast suspension to a final concentration of 500 nM. Fluorescence was observed with a Leica TCS SP5 confocal laser scanning microscope. Subcellular localization of *Chlamydomonas* HCS1 and HCS2 was predicted using WoLF PSORT (https://wolfpsort.hgc.jp/) and PredAlgo (http://lobosphaera.ibpc.fr/predalgo)[78], a specific program dedicated to the prediction of protein localization to the nucleus, chloroplast or mitochondrion in green algae with high confidence.

### Yeast complementation assay

The yeast strains Y00367 (BY4741; *MATa*; *ura3*Δ0; *leu2*Δ0; *his3*Δ1; YAL039c::kanMX4) and Y04936 (BY4741; *MATa*; *ura3*Δ0; *leu2*Δ0; *his3*Δ1; YKL087c::kanMX4), which contain a knock-out of the gene encoding either holocytochrome *c* synthase (CCHLp, Δ*cyc3*) or holocytochrome *c*$_1$ synthase (CC$_1$HLp, Δ*cyt2*), were ordered from Euroscarf (http://www.euroscarf.de). Both strains were transformed with a plasmid containing either the *C. richardii HCCS* (*CriHCCS*) full coding sequence or an empty vector lacZ under the control of the 3-Phosphoglycerate kinase 1 (PGK1) promoter. Both vectors were assembled using Golden Gate cloning techniques[74,75]. Two similar vectors were assembled by replacing the *CriHCCS* sequence with the full-length coding sequence of *C. reinhardtii HCS1* (*CreHCS1*) or *HCS2* (*CreHCS2*), and these vectors were also transformed into the Δ*cyc3* and Δ*cyt2* strains. Transformants were selected on solid synthetic defined (SD) media consisting of 0.67% (w/v) yeast nitrogen base without amino acids, supplemented with yeast synthetic dropout medium without Leucine (SD-Leu). For complementation tests, constructed yeast strains were grown on SD-Leu supplemented with either 2% (w/v) glucose or 3% (w/v) glycerol. Primer sequences used for making these constructs are provided in Supplementary Data 3.

### *Chlamydomonas* strains and culture conditions

*C. reinhardtii* strain g1 (CC-5415; nit1, mt$^+$) was used as the background strain for gene editing experiments. Unless noted otherwise, cultures were incubated under continuous illumination (150 μmol m$^{-2}$ s$^{-1}$) on an orbital shaker (180 rpm) at 25 °C. For transformation, an initial batch of cells was pre-cultured, after inoculation from one-week-old Tris-Acetate-Phosphate (TAP) agar plates, to an optical density of ~0.4 at 750 nm in TAP medium. A one-tenth aliquot was then transferred into fresh TAP medium supplemented with 1 μg mL$^{-1}$ cyanocobalamin (vitamin B$_{12}$), to enhance *Chlamydomonas* thermal tolerance. Prior to electroporation, cells were collected at the exponential growth phase (~2 × 10$^6$ cells mL$^{-1}$) by centrifugation at 3000 × g and resuspended in TAP containing 40 mM sucrose and 1 μg mL$^{-1}$ cyanocobalamin to a final density of ~2.2 × 10$^8$ cells mL$^{-1}$.

### CRISPR/Cas9 RNPs, transformation and screening

We used the commercially available Alt-R CRISPR-Cas9 System developed by Integrated DNA Technologies (IDT) for gene editing. The 20-nt guide sequence, corresponding to the target-specific protospacer region was designed using the online tool CHOPCHOP v2 (https://chopchop.cbu.uib.no)[79]. *C. reinhardtii* transformation was carried out by electroporation of the CRISPR/CAS9 RNP complex and single-stranded DNA that targets precise editing of the HCS1 or HCS2 locus[80]. To aid selection, the paromomycin resistance cassette (aphVIII transgene) amplified by high-fidelity PCR from the pSI103 plasmid[81] was co-transfected into cells by electroporation. Electroporated cells diluted in 500 μL TAP medium containing 40 mM sucrose and 1 μg mL$^{-1}$ cyanocobalamin were placed on an orbital shaker (50 rpm) under light (50 μmol m$^{-2}$ s$^{-1}$) for 2 h at room temperature. Cells were then heat shocked at 39 °C for 30 min in a water bath and placed back to the orbital shaker for a 36 h recovery. Cells from each electroporation

were spread on two TAP agar plates containing 10 µg mL$^{-1}$ paromomycin for selection. The plates were incubated at room temperature under continuous illumination (100 µmol m$^{-2}$ s$^{-1}$) until visible colonies appeared. About 100–150 colonies were randomly selected and screened with the Chelex colony PCR method[82]. To facilitate the screening process, we also transferred the selected colonies to fresh TAP plates and then kept the plates in the dark for a week. Colonies with reduced or no growth in the dark were chosen as priority for PCR screening. The oligonucleotides and primers used for these experiments are listed in Supplementary Data 3.

### Measurement of *Chlamydomonas* growth and respiration

For growth curve measurement, cells were pre-cultured as described above and then inoculated to TAP medium. Three replicates of each strain were cultivated under continuous light or in the dark. Culture growth was examined by measuring daily absorbance at 750 nm over a period of 6 days. Ten-fold dilution spot test was conducted by spotting 10 µL of cell suspension (at an initial concentration of $10^3$–$10^4$ cells/µL) on TAP agar plates and incubating the plates under light for 3 days or in the dark for 8 days.

Colorless TTC (2,3,5-triphenyltetrazolium chloride) turns red when under reducing conditions and has been used as a redox indicator of mitochondrial ETC functionality in microorganisms[83,84]. Our TTC staining for *C. reinhardtii* was performed on TAP agar plates (9 cm in diameter) overlaid with 15 mL agar medium containing 0.5 mg/mL TTC[50]. The plates were incubated at 30 °C for 5 h in the dark and then photographed.

Measurement of whole-cell oxygen consumption was carried out in a dark room using a Clark Electrode (Oxygraph, Hansatech Instruments)[85]. To assess the activity of mitochondrial complexes, sodium cyanide (NaCN) and salicylhydroxamic acid (SHAM) were used as specific inhibitors and injected from the top hole of the electrode chamber to cell cultures at a final concentration of 0.5 mM. Respiration rates were recorded for 3 min on each sample before and after the addition of each inhibitor. All measurements were conducted in triplicate with the same setup.

### Metabolite profiling of *Chlamydomonas* HCS knock-down mutant

*Chlamydomonas* cells were pre-cultured as described above and then transferred into fresh TAP medium and grown to middle logarithmic phase (~2 × 10$^6$ cells mL$^{-1}$) under two continuous light conditions at 150 µmol·m$^{-2}$·s$^{-1}$ (normal light) and 50 µmol m$^{-2}$ s$^{-1}$ (low light). Cells were harvested by centrifugation at 4000 × $g$ for 1 min, washed twice, and resuspended in ultra-pure water to an OD$_{750}$ of 1.0. Cells in one milliliter of the cell suspension were collected by vacuum filtration through a membrane filter (0.45 µm HV Durapore 25 mm diameter; Millipore Sigma, Burlington, MA, USA). The filter was rolled up and placed into a 2 mL microcentrifuge tube, frozen in liquid nitrogen, and stored at −80 °C. Cellular metabolites were extracted according to Obata, Schoenefeld[86] with slight modifications. The filter was soaked in 700 µL methanol supplemented with 5 µL of 0.2 mg/mL ribitol solution and vigorously vortexed for 15 s. Following 15 min incubation at 70 °C with shaking, the filter was removed, the solution was centrifuged at 17,500 × $g$ for 10 min, and the supernatant was mixed with 750 µL water and 325 µL chloroform by vortexing. After centrifugation at 17,500 × $g$ for 10 min, 400 µL of the upper aqueous phase was taken into a new microcentrifuge tube and dried down in a centrifugal vacuum concentrator (CentriVap, Labconco, Kansas City, MO, USA). Dried metabolites were dissolved into 20 mg/mL methoxyamine hydrochloride solution in pyridine and incubated for 2 h at 37 °C with shaking for methoxyamination. The metabolites were further trimethylsylilated by incubating for 30 min at 37 °C with shaking after the addition of 70 µL N-Methyl-N-(trimethylsilyl) trifluoroacetamide (Millipore Sigma). Gas chromatography-mass spectrometry (GC-MS) data

acquisition, peak annotation, and quantification were conducted[87], and the peak heights of the representative ions for each metabolite peak were normalized by that of internal standard (ribitol) to represent relative levels of metabolites. The parameters used for the peak annotation are provided in Supplementary Data 1.

### Functional complementation of Arabidopsis *ccmh* line

Seeds for a *ccmh* (At1g15220) T-DNA line in the Arabidopsis (*Arabidopsis thaliana*) Columbia-0 background were obtained from the Arabidopsis Biological Resource Center (Ohio State University, Columbus, OH). The insertional *ccmh* line (SALK_046872) was genotyped by PCR using the primers listed in Supplementary Data 3 and confirmed by sequencing.

A codon optimized *CriHCCS* CDS sequence (system III) was synthesized (GenScript, USA) and assembled downstream of the Ubiquitin 10 promoter to create the vector *pUBQ10::CriHCCS*. The constructed plasmids were then transformed into viable heterozygous *ccmh/CCMH* plants by the floral dip method via *Agrobacterium tumefaciens* strain GV3101. Homozygous *ccmh* plants due to complementation were isolated by selecting seeds on plates containing 10 µM glufosinate ammonium (BASTA, for transformant selection) and PCR genotyping. Progenies from independent transformants were verified for *ccmh* T-DNA homozygosity and *CriHCCS* presence by PCR genotyping with the primers listed in Supplementary Data 3.

Seeds collected from the T3 complemented lines and the WT plants were surface-sterilized (70% v/v ethanol and 50% bleach) and sown on Murashige and Skoog (MS) agar plates (half strength MS medium, 0.05% w/v MES and 0.75 w/v agar, pH 5.8) followed by stratification at 4 °C in the dark for 3 days. One-week-old seedlings were transferred to soil and grown under controlled long-day conditions (16 h light/8 h dark, 22 °C, 50% relative humidity) for growth-stage-based phenotypic observation. Ten days after flowering, siliques were dissected using a fine tweezer and visualized under a stereomicroscope (SMZ25, Nikon).

Seed germination test was conducted by transferring the MS plates after cold stratification to light (100 µmol m$^{-2}$ s$^{-1}$) at 22–24 °C. Germination rate was scored as the percentage of seeds with visible radicle protrusion at each time point. Approximately 100 seeds were used for each line. Seedlings from two-days-old germination plates were transferred to the surface of new MS plates and placed vertically in a growth chamber (16 h light and 8 h dark period with approximately 100 µmol m$^{-2}$ s$^{-1}$ illumination at 22 °C. Root length was measured at day 14 after seed sowing. Both experiments were repeated three times.

### Phylogenetic analysis of eukaryotic HCSS homologs

Eukaryotic HCCS homologs (accession numbers listed in Supplementary Fig. 3) were identified by using Archaeplastida HCCS sequences (Supplementary Table 2) in an online blastp search (https://blast.ncbi.nlm.nih.gov/Blast.cgi) against the NCBI non-redundant protein database and in a tblastn v2.2.31 search against newly assembled transcriptomes (Supplementary Data 2). HCCS proteins were aligned using MAFFT version 7.245[88] with the most accurate 'linsi' setting. Alignments were trimmed to remove poorly aligned regions using trimAl v1.2[89] with the trimgappy setting. Maximum-likelihood trees were built from the trimmed alignment with RAxML v8.2.4[90] using the PROTGAMMAAUTO model option, and branch support was assessed with 1000 rapid bootstrap replicates. Bayesian inference trees were constructed with MrBayes v3.2.7[91] using default priors and two independent runs, each consisting of one cold and three heated chains. Runs were completed after 5 million generations, with trees sampled every 1000 generations and the first 25% of the sampled trees discarded as burn-in. AU topology tests were performed with IQ-TREE v2.2.2.6[92] by comparing the ML tree to alternatively constrained trees.

**Reporting summary**

Further information on research design is available in the Nature Portfolio Reporting Summary linked to this article.

## Data availability

Sequence data generated in this study have been deposited in GenBank under accession BK063718 -BK063750 and the NCBI SRA under accession PRJNA976711). Source data are provided with this paper.

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

## Acknowledgements

The authors thank Wenhu Guo for performing the initial analysis of the diversity of systems I and III; Sally Cheung, Samantha Link, Leticia Pasqualino, Brittany Schweiger and Trisha Vickrey for cultivation and maintenance of ferns and Arabidopsis lines in the lab and greenhouse; Nick Amendola and Rebecca Roston for assistance with CriHCCS plasmid construction; Géraldine Bonnard for informative discussions on CCM systems; Bara Altartouri for assistance with Arabidopsis silique dissection and imaging; Ciera Haynes and Geovanna Sicoli Carrasquero for helping with Arabidopsis genotyping; and Iryna Bohovych, Oleh Khalimonchuk and Yoshihiro Shiraiwa for providing the Oxygraph instrument and useful discussions about mitochondrial assays. The research was supported by the National Science Foundation (Award MCB 2212075 to J.P.M.) and the University of Nebraska (Collaboration Initiative grant 19501 to J.P.M. and H.C.).

## Author contributions

J.P.M. conceived the project and acquired project funding. H.L., S.A., C.E., J.J.O., M.R., T.O., C.C. and J.P.M. performed experiments and analyses. H.L., S.A., T.O., C.C., H.C. and J.P.M. interpreted results. H.L. and J.P.M. wrote the manuscript, and all authors contributed to revising the manuscript.

## Competing interests

The authors declare no competing interests.
