## [Peer Review File · Nature Communications]

Recurrent Evolutionary Switches of Mitochondrial Cytochrome c Maturation Systems in ArchaeplastidaReviewers' Comments:

Reviewer #1:

Remarks to the Author:

This manuscript presents some interesting phylogenetic patterns in terms of gene losses/retention and complements it with experimental functional studies. In that sense it is a very interesting study that draws attention to an interesting evolutionary outcome being played out across diverse photosynthesizing lineages.

A few concerns come to mind as I read the manuscript:

1) Lines 122-144 (and elsewhere): Loss of ccm genes assumes they are not too divergent to identify by standard means (BLAST). In plant mt DNA, there are variable rates of nucleotide substitution and if rate-mutant lineages are examined their genes are often very difficult to identify/compare with other more conserved sequences. The fact that very anciently divergent lineages are being compared could exacerbate this potential problem. At the very least, I recommend that the authors determine/mention if any of these lineages examined and apparently missing ccm genes are "normal" for their other genes. That way, it would alleviate any concerns about false negatives. Afterall, proving that something does not exist is difficult.

2) Lines 156-157 and throughout: The authors state that a gain of system III has occurred. There is nothing about any of the patterns shown that make me think system III has been gained by horizontal gene transfer or any other process. It looks to me that both system 1 and III have co-occurred for a long period of time but system I has been lost since it was apparently redundant with system III. And, for other plants, the opposite has occurred in which no system III is known but system I is in place (all seed plants?). Nonetheless, the numerous and recurrent losses of system I is remarkable and interesting.

3) Fig. 6 & lines 339-347 & 405-420: The authors state: Within land plants, however, many HCCS relationships do not agree with organismal relationships(Figure 6B). In particular, ferns and lycophytes do not form expected monophyletic groups. Instead, *Ophioglossum* (a fern) is nested within a clade for *Selaginella* (a lycophyte), whereas *Isoetes* (a lycophyte) is nested within a clade of leptosporangiate ferns from Osmundales, Gleicheniales and Hymenophyllales, and the other leptosporangiate ferns from Polypodiales are sister to all other land plants. These nonmonophyletic relationships suggest a history of horizontal transfer rather than vertical inheritance for HCCS from some ferns and lycophytes.

The main problem with this section is that none of the putative horizontal transfers inferred from the phylogenetic relationships is well-supported by bootstrap values. In fact, 68 is the highest shown for *Ophioglossum* nested within *Selaginella*. *Isoetes* within ferns is even weaker at 51. None of these conclusions are supported in a statistical sense so the inferences cannot be claimed with any confidence.

4) Lines 444+: Because there is a lack of convincing evidence for horizontal gene transfer of HCCS, I am not even sure this is a case of convergent evolution. Rather, if the ancestor of all green plants had both system I and system III, then there has been stochastic loss of one or the other in numerous lineages. A stochastic loss is not necessarily adaptive which questions whether this is convergence. The authors seem to highlight the loss of system I with retention of system III but the opposite is also fascinating and interesting: the loss of system III and retention of the seemingly more complex system I.

Reviewer #2:

Remarks to the Author:

In this paper, Li and colleagues examined evolution of genes involved in CCM systems I and III among Archaeplastida species. They found that the loss of system I is compensated by the gain of system III. In addition, they found that genes HCCS from system III can rescue yeast HCCS deletion lines and they also validated the functions of genes HCCS in the model green alga. Lastly, they examined the distribution of genes HCCS across eukaryotic organisms.

Overall, the work is well written, but the pieces of evidence are insufficient to explain the loss and gain for systems I and III. Here I have some concerns that authors might wish to address.

1) Regarding the presence-absence of genes involved in systems I and III among Archaeplastida species, they used the representative system I and III genes from *Arabidopsis thaliana* to screen the genomes and transcriptomes. This analysis is oversimplified. Three issues might lead to the pattern found in Figure 1. First, taking the representative system I and III genes as queries is not reasonable. It would be good to use all genes in systems I and III; Second, the taxon samplings included in this study are insufficient; you should screen all high-quality Archaeplastida genomes. Sparse taxon samplings definitely influence your conclusion of the loss and gain for systems I and III. Third, given that species you examined are highly diverse, using the blast search to identify homologs of system I and III genes could be problematic. You might first build hidden markov model on representative species (each can be selected for each clade) and then search the homologs of the system I and III genes.

2) In the main text, you mostly stated the system III. The system III can rescue yeast HCCS deletion lines, which is good. How about the systems I? Can systems I rescue yeast HCCS deletion lines? If not, it's problematic to come with a conclusion, which is that Archaeplastida lineages independently switched from system I to III

3) The section "Phylogenetic analysis of eukaryotic HCCS homologs" might be unnecessary. If you really want to include it, you might need to reanalyze this. you could use the above approach (see concern #1) to screen system I and III genes in the genomes of species in major eukaryotic lineage (not all eukaryotic genomes), rather than in NCBI non-redundant protein database.

Reviewer #3:

Remarks to the Author:

The manuscript reports the analysis on CCM system I and III. The authors comprehensively surveyed the presence/absence of CCM system I and II genes across diverse eukaryotes and found that sparsely distributed pattern of CCM system III correlates with the loss of CCM system I. They confirmed that putative system III homologs in viridiplantae indeed localize in mitochondria and is sufficient to complement in system III deficient yeast. Furthermore, using CRISPR-Cas9 system, they generated *Chlamydomonas* system III mutants and proved effects of system III on respiration. The authors performed phylogenetic analysis and proposed that both system I and III had existed in the archaeplastid common ancestor, but one of them were eventually lost, and several independent HGTs of system III within some land plant lineages followed after that. The manuscript is well written and the study is very interesting in that it reveals the distribution of CCM genes across the eukaryotes and discussed about the implications. However, fundamentally I am doubting the novelty of this study, as the critical results and discussions in the manuscript had already been reviewed in Allen, J.W.A., Jackson, A.P., Rigden, D.J., Willis, A.C., Ferguson, S.J. and Ginger, M.L. (2008), Order within a mosaic distribution of mitochondrial c-type cytochrome biogenesis systems?. The FEBS Journal, 275: 2385-2402. <https://doi.org/10.1111/j.1742-4658.2008.06380.x>.

First, Allen et al. did a comparative genomics analysis and found that system I and system III are not compatible and system I genes are always bipartite into mitochondrial and nuclear genome. This statement is identical to the finding of the authors' study: mosaic distribution of CCM pathway. Second, Allen et al. performed a phylogenetic analysis of system III genes (heme lyases). Though with less taxa, different topology and the extent of lineages, development of the reasonings in both Allen et al and this manuscript resemble each other. Allen et al proposed two models: 1) origin in common ancestor of bikont and unikont, or 2) multiple LGT of system III gene (heme lyase), and concluded

that model 2 is more likely to have occurred, unlike the author's statement that both systems presented in the Archaeplastida common ancestor (which resembles model 1). Third, Allen et al. discussed possible reasons that could explain the preference toward system III over system I. Here, the content is slightly different, but essentially, both discuss the same thing. Allen et al. proposed that no need of wide substrate specificity (as only need to mature two mitochondrial cytochromes c) gave selective force for system III. On the other hand, the authors said that complexity of the system I made simpler system III to thrive in most eukaryotes. However, those that still carry system I would benefit from enhanced regulatory control over the mitochondrion. Allen et al. also mentioned similarity of between CCM system I&III and EF-1 α and EFL. The authors did a fine work to experimentally prove the function of putative system III genes in green algae. However, the results are not so much surprising because long have been known that the genes are system III genes. Though putative, the genes grouped well with experimentally proven system III genes in the phylogenetic tree and distinct substrate preferences of system III genes were already studied. In conclusion, the experiments the authors did were novel and appreciable, but these alone is not enough. The authors need to provide and clarify their novelty given their findings were already studied before.

Other than that, I have few minor comments.

1. Line 42-43 "Convergent evolution describes the independent evolution of a similar trait in different species": Add reference?
2. Line 77-80: Provided references seem to cover only for plants, not eukaryotes.
3. Line 80-84: is an interesting point for readers who are not familiar with ccm systems. Do a single system III gene perform all the roles of the eight system I genes? It would be helpful for readers if the functions of system III genes are described with the comparison to system I genes.
4. Line 156-157: Then who would be the donor?
5. Line 286-289: Please describe shortly why there are no differences under normal light conditions.
6. Line 315: BS value 87 is not a low value but also, neither is strong. I would not say it is "strong".
7. Line 345-347: Proposed HGT is not convincing as single gene phylogeny often partially fail to correctly reconstruct phylogenetic relationships. The authors should remove statements regarding HGT within streptophytes, or provide more evidences.
8. Line 459-462: There are many other lineages that carries system I. Will the same reasoning apply to the lineages other than seed plants? Or is it just simply because those lineages have never experienced HGT? Also, references that system I enhances mitochondrion control should be provided.
9. Figure 5: What does red arrows stand for?
10. Figure 6 (also sup. fig. 3): It should be mentioned that BS value under 50 is not shown. Also, how is the tree rooted?
11. Data availability: Alignments and treefiles must be provided.
12. Sup. Fig. 3: BS value 39 was not removed (see XP_005847167).

Reviewer #4:

Remarks to the Author:

It remained unclear why and how mitochondrial CCM cytochrome maturation system I have been replaced by HCCS-mediated CCM cytochrome maturation system III multiple times independently in eukaryotic evolution. Li et al. tackled this issue by focusing on Archaeplastida. They re-investigated the phylogenetic distribution of system I and III in Archaeplastida and in eukaryotes. They found at least 11 independent evolutionary losses of system I genes in Archaeplastida. They also demonstrated that Ceratopteris and Chlamydomonas HCCSs could complement functions of the yeast homologue, indicating that the green HCCSs are indeed cytochrome synthases. On the basis of the implication of the phylogenetic analysis, the authors claimed that independent lateral gene transfers have shaped the current phylogenetic distribution of CCM and HCCS in Archaeplastida. The current work includes important clues to gain insight into functions and evolution of CCM and HCCS. However, all the

biochemical, genetic, and phylogenetic analyses seem indirect or incomplete for the purpose. In addition, Discussion is comprised of large parts independent from data or evidence. I therefore suggest additional experiments and analyses.

First, complementation of the yeast HCCS mutants by *Chlamydomonas* HCCS is the important step for clarifying HCCS functions and evolution in eukaryotes. However, the purpose of this study is to understand why and how CCM system I genes have been replaced by system III HCCS genes. If so, complementation of the CCM mutant of *Archaeplastida* (e.g., *Arabidopsis*) by a HCCS should be investigated. Whether CCM can be complemented by a HCCS and whether the complementation does not exhibit any growth impairment might provide insight into evolution of CCM/HCCS replacement. Relevant to the above, it is not clear whether the authors have evaluated or decreased the possibility of off-target in the CRISPR-Cas9 experiment.

Second, as mentioned by the authors, the phylogenetic tree is unresolved. Therefore, it is difficult to interpret anything from the tree.

Accordingly, the current Discussion is comprised of large part of assumption not derived from any data. For instance, lines 401-402 mention that "the two systems are essentially incompatible." I could not find any evidence, references, or data for this. If the authors would like to suggest it, they should establish a transformant plant having both systems and investigate its growth and whether cytochrome maturation is impaired by the coexistence. The same transformant would also allow to investigate the authors' suggestions in lines 458-459, on benefit of system III.

In this point of view, we should not forget that any gene transfer events and thereby gene replacements always follow coexistence of an endogenous gene and an exogenous transferred gene for a certain evolutionary period, prior to loss of the endogenous one. This is because the endogenous gene cannot be non-functionalized or lost prior to functionalization of the transferred gene for the cell viability. Thus, both systems should have functioned in a same cell although it is difficult to imagine and evaluate how long they have coexisted. Gene transfer events of HCCS might imply that coexistence of systems I and III are not incompatible for a certain term.

Further, in line 406-420, since the tree is not resolved well, the tree topology does not provide any clue to gain insight into evolution of HCCS. So are lines 429-435. For the discussion, the authors could statistically evaluate whether the alternative tree showing each of ferns, lycophytes, and hornworts is monophyletic is rejected or not by the AU test. If rejected, the authors' scenario might in part be supported.

lines 156-157

I could not follow this. The current distribution can also be explained by differential losses after coexistence of system I and III by which "extant species" have completely lost either of them. I do not say the authors' claim and the evolutionary scenario are wrong. But I would say the authors' claim is not based on any data and therefore which evolutionary scenario is more likely cannot be evaluated quantitatively. Currently, both scenarios are not rejected.

Fig1 and Fig6 are inconsistent in light of taxon sampling. Some species included in Fig6 are not in fig.1 such as *Dicranopteris*. Since Fig.1 is the source data for Fig. 6, they should be consistent to each other.

Reviewer #1 (Remarks to the Author):

This manuscript presents some interesting phylogenetic patterns in terms of gene losses/retention and complements it with experimental functional studies. In that sense it is a very interesting study that draws attention to an interesting evolutionary outcome being played out across diverse photosynthesizing lineages.

A few concerns come to mind as I read the manuscript:

1) Lines 122-144 (and elsewhere): Loss of *ccm* genes assumes they are not too divergent to identify by standard means (BLAST). In plant mt DNA, there are variable rates of nucleotide substitution and if rate-mutant lineages are examined their genes are often very difficult to identify/compare with other more conserved sequences. The fact that very anciently divergent lineages are being compared could exacerbate this potential problem. At the very least, I recommend that the authors determine/mention if any of these lineages examined and apparently missing *ccm* genes are "normal" for their other genes. That way, it would alleviate any concerns about false negatives. After all, proving that something does not exist is difficult.

We thank the reviewer's valuable feedback. Here, we understand the concern about the difficulty in proving a negative. For the survey of system I, we previously used *Arabidopsis* system I homologs in the blast searches. Whereas for the system III HCCS survey, we used HCCS from *Chlamydomonas* (green alga), *Selaginella* (land plant) and *Chondrus* (red alga) as queries in the full search, which improved the ability to detect diverse homologs. To increase consistency between the system I and III surveys, and to improve search approach for system I survey, we now performed the system I survey using homologs from diverse queries: *Arabidopsis* (land plant), *Klebsormidium* (green alga) and *Cyanidioschyzon* (red alga). This had no effect on our results. We updated our methods to reflect this change (lines 545-549) and added a new supplementary table S3 to list the query sequences used.

For the survey in Figure 1, we should point out that we have several layers of concordance that support our findings. First, the presence/absence of system I is consistent between the nuclear encoded components of system I (CcmA, E, H), which is based on blasting nuclear genomes/transcriptomes, and the mitochondrial encoded components of system I (ccmB, C, F), which is based on blasting complete mitogenomes. Second, we specifically chose to include species that had a sequenced mitogenome, which allowed us to corroborate the presence/absence of mitochondrial genes between our blast survey and the available mitogenome annotation information. To make this clearer, we have added a statement in the results (lines 144-151) to discuss the issue of the reliability of our results, based on the importance of available mitogenomes, and the fact that our gene survey results are corroborated by (and expand upon) results of previous studies and by the concordance between the mitogenome and nuclear genome/transcriptome data. As a side note, in the previous version of figure 1, *Interfilum* was included despite lacking an available mitogenome, resulting in uncertainty regarding the status of its *ccmB* gene. To highlight the importance of having an available mitogenome for the survey in Fig. 1, we chose to remove *Interfilum* from this figure. At the same time, in response to other reviewer comments, we have added seven additional species (*Amborella*, *Thuja*, *Azolla*, *Raphidocelis*, *Chloropicon*, *Pycnococcus*, *Porphyridium*) which have nuclear genome data.

With respect to evolutionary rates, most plant mitochondrial genomes are very slowly evolving. In green algae, mitochondrial and plastid rates are generally similar, while in red algae the mitochondrial rates are higher than in the plastid. Nevertheless, mitochondrial genes, including the *ccm* genes, are easily detectable (if present) by blast searches, especially when using diverse queries, as we have now done. We are quite familiar with the rare high-rate plant mitochondrial lineages (eg angiosperms such as *Plantago*, *Pelargonium*, *Silene* and *Viscum*, as well as the red alga *Galdieria sulphuraria* and green algae in Chlamydomadales), as my lab has been involved in the characterization of many of these high-rate lineages.

- a) Among angiosperms, we avoided all fast-rate species simply because we focused on better known angiosperm taxa (such as *Arabidopsis* and rice), whose mitogenomes are normally slow.
- b) Among chlorophytes, we did include *Chlamydomonas*, which has fast-evolving mitochondrial genes, because it is one of our focal species of interest in the manuscript. The *Chlamydomonas* mitogenome lacks any annotated system I genes, and it is exceedingly unlikely that four previously unrecognized genes (totaling ~5 kb in length) could be present in this very small (~16 kb) mitogenome. More generally, the system I gene annotations are absent from all of the >100 chlorophytic green algal mitogenomes available in GenBank. Most of these species are slowly evolving. Overall, we feel there is essentially no chance we (and other scientists) missed these system I genes from chlorophytes, regardless of the rate of mtDNA evolution.
- c) The only other obviously fast-evolving mitochondrial lineage included is *Galdieria*. It is an interesting example because we indeed have some uncertainty about the status of *ccmB* (as noted in Fig. 1). In our publication on this mitogenome (Jain et al 2015 GBE 7: 367-380), we could not find an obvious candidate in the mitogenome or the nuclear genome, which is a consequence of blast limitations due to its fast rate of mitogenome evolution and its very distant relationship to any other red algal species. However, we could confidently identify all of the other system I CCM genes in this fast-evolving species. The fact that we could identify nearly all of the system I CCM genes, even in this highly divergent species, provides fairly compelling evidence that our approach is reliable.

2) Lines 156-157 and throughout: The authors state that a gain of system III has occurred. There is nothing about any of the patterns shown that make me think system III has been gained by horizontal gene transfer or any other process. It looks to me that both system I and III have co-occurred for a long period of time but system I has been lost since it was apparently redundant with system III. And, for other plants, the opposite has occurred in which no system III is known but system I is in place (all seed plants?). Nonetheless, the numerous and recurrent losses of system I is remarkable and interesting.

We agree with the reviewer that our wording in the abstract, introduction and results should be more neutral about the origin and evolution of system III. We have adjusted our statements in multiple places in the abstract, introduction and results to talk about the presence/absence rather than gain/loss of the two systems.

In the discussion, we present two different possibilities to explain the variable presence of system I and III: 1) Ancestral presence of both systems and repeated stochastic loss of one or the other, or 2) rampant HGT of system III followed by stochastic loss of one or the other (where HGT would only be detected if system III was retained). We have made extensive efforts to impress upon the reader that both hypotheses have complications that make them unlikely. The main argument against the long-term co-

occurrence of both systems is the fact that we find essentially no eukaryotes that use both today. To date, only 2 of 7 species within the Provora appear to have both systems. However, there is a lack of experimental evidence that both systems are functional, and sequence contamination is a possibility. If both systems have been compatible and maintainable for hundreds of millions of years, then why don't we find more species that continue to use both? While acknowledging the reviewer's perspective that the evidence for Horizontal Gene Transfer (HGT) is not strong, we have taken steps to address this concern. As described below, we have implemented additional analyses to increase the evidence for phylogenetic incongruence that suggests HGT.

3) Fig. 6 & lines 339-347 & 405-420: The authors state: Within land plants, however, many HCCS relationships do not agree with organismal relationships (Figure 6B). In particular, ferns and lycophytes do not form expected monophyletic groups. Instead, *Ophioglossum* (a fern) is nested within a clade for *Selaginella* (a lycophyte), whereas *Isoetes* (a lycophyte) is nested within a clade of leptosporangiate ferns from Osmundales, Gleicheniales and Hymenophyllales, and the other leptosporangiate ferns from Polypodiales are sister to all other land plants. These nonmonophyletic relationships suggest a history of horizontal transfer rather than vertical inheritance for HCCS from some ferns and lycophytes.

The main problem with this section is that none of the putative horizontal transfers inferred from the phylogenetic relationships is well-supported by bootstrap values. In fact, 68 is the highest shown for *Ophioglossum* nested within *Selaginella*. *Isoetes* within ferns is even weaker at 51. None of these conclusions are supported in a statistical sense so the inferences cannot be claimed with any confidence.

We agree with the reviewer that the bootstrap values are not extremely convincing. However, this is not too surprising as we are doing deep phylogenetics with a single and fairly small (~300 AA) HCCS protein. For Figure 6B, we added three more taxa to the phylogeny (the fern *Azolla*, the liverwort *Treubia*, the red alga *Rhodospirillum rubrum*). We also updated the alignment and trimming procedure to increase the useful characters. To provide additional statistical analysis beyond ML bootstrapping, we performed Bayesian phylogenetic analysis (shown in Figures 7 and S3B), which provides a more statistically interpretable result relative to bootstrapping. These updated ML and Bayesian results provide stronger bootstrap support and statistically significant (>0.95) Bayesian posterior probabilities for the anomalous position of *Ophioglossum* within *Selaginella*. We also performed topology tests using the Approximately Unbiased (AU) method, as implemented in IQ-TREE. These topology tests statistically reject the possibility of fern and/or lycophyte monophyly. See relevant text in the results (lines 412-415). Overall, we now provide statistical support for the unusual relationships among land plants, which is consistent with a history of HGT rather than vertical inheritance.

4) Lines 444+: Because there is a lack of convincing evidence for horizontal gene transfer of HCCS, I am not even sure this is a case of convergent evolution. Rather, if the ancestor of all green plants had both system I and system III, then there has been stochastic loss of one or the other in numerous lineages. A stochastic loss is not necessarily adaptive which questions whether this is convergence. The authors seem to highlight the loss of system I with retention of system III but the opposite is also fascinating and interesting: the loss of system III and retention of the seemingly more complex system I.

We appreciate the reviewer for this insightful criticism. Yes, we agree that there are two different possibilities to explain the disjunct distribution of HCCS among Archaeplastida: Ancestral presence and stochastic loss, or rampant HGT. While the evidence for HGT may be limited, we should also point out that there is no evidence whatsoever suggesting that the ancestor of all Archaeplastida had both systems; this is simply an inference that has its own concerns because it requires that both systems, which would be redundant, to have nevertheless been maintained for hundreds of millions of years, only for all lineages to have decided at some point to dispense with one or the other prior to present day.

Whether this is convergent evolution or not (regardless of the underlying process that leads to system I to III shifts) touches on the broader issue in the literature of the differing ways to define convergent evolution. Several of the citations in the introduction dive into this issue in detail, describing the different ways (process based vs. phenotype based vs. adaptation based) in which convergent evolution is discussed in the literature. We strongly disagree that convergent evolution must be adaptive. Instead, we feel that our results provide a clear mechanistic way in which convergent evolution may arise non-adaptively. Nevertheless, we do not wish to wade into this broader discussion of the shifting and ambiguous definitions of convergent evolution, so we have modified the text throughout to refer more generally to repeated evolution, rather than to convergent evolution.

Reviewer #2 (Remarks to the Author):

In this paper, Li and colleagues examined evolution of genes involved in CCM systems I and III among Archaeplastida species. They found that the loss of system I is compensated by the gain of system III. In addition, they found that genes HCCS from system III can rescue yeast HCCS deletion lines and they also validated the functions of genes HCCS in the model green alga. Lastly, they examined the distribution of genes HCCS across eukaryotic organisms.

Overall, the work is well written, but the pieces of evidence are insufficient to explain the loss and gain for systems I and III. Here I have some concerns that authors might wish to address.

We appreciate the reviewer's concise summary of the key achievements in this work and the constructive feedback for improvement.

1) Regarding the presence-absence of genes involved in systems I and III among Archaeplastida species, they used the representative system I and III genes from *Arabidopsis thaliana* to screen the genomes and transcriptomes. This analysis is oversimplified. Three issues might lead to the pattern found in Figure 1. First, taking the representative system I and III genes as quires is not reasonable. It would be good to use all genes in systems I and III;

We apologize for the confusion in our wording. We did in fact use all genes for system I (ccmA, B, C, E, F, H) and system III (HCCS only) in our survey; our use of the term “representative” was meant to indicate that we collected genes from a few species representatives. This sentence (lines 545-549) was updated not only to clarify this issue by removing the term “representative”, but also to accommodate the inclusion of more species representatives in response to other reviewers’ concerns.

Second, the taxon samplings included in this study are insufficient; you should screen all high-quality Archaeplastida genomes. Sparse taxon samplings definitely influence your conclusion of the loss and gain for systems I and III.

With respect to taxon sampling in figure 1, we respectfully disagree that our sampling is sparse or insufficient. We also disagree that adding additional genomes would definitely influence our results. For the nuclear gene component of the survey, we already sampled a diverse collection of nuclear genomes that were available. While we could certainly add more species, the vast majority of these unsampled nuclear genomes are from angiosperms and chlorophytes, but the status of their usage of system I or III is not at all in question. We can see system I genes are clearly present in all angiosperm mitogenomes, and we can clearly see the lack of system I genes in all chlorophyte mitogenomes. Adding more of these taxa does not provide any additional information, so for brevity of presentation in the figure, we don't think more taxa are needed.

Outside of angiosperms and chlorophytes, there are relatively few species with an available mitogenome and a high-quality nuclear genome. As described in detail in response to reviewer 1, our first criterion was to ensure that the species sampled in Figure 1 had a mitogenome, which ensured we could be very confident about our mitochondrial presence/absence calls. For some of the streptophytic green algae with nuclear genomes (eg, *Chara*, *Mesostigma*, *Chlorokybus*), essentially none of them are "high-quality" genomes (certainly not chromosome-scale which is the standard today). Because of their fragmentation, annotations of these genomes can be unreliable, so we preferred to use our own assembled transcriptome, which are at least as good, especially because these system I and III genes are essential, undoubtedly expressed in essentially all tissues.

Indeed, our choice to build transcriptomes (rather than relying on nuclear genomes only) enabled a much more diverse collection of species, with a particular focus of phylogenetically diverse groups that are generally underrepresented in nuclear genome and mitogenome sequencing (eg, ferns, lycophytes, streptophytic green algae). This choice in fact provides a much richer set of taxa than could be collected if we chose to use nuclear genomes only, and it avoids any annotation issues of nuclear genomes completely because we can survey mature transcript sequences directly. So we respectfully disagree that our taxon sampling is sparse or insufficient.

Nevertheless, there were certainly additional taxa that we could add to the Figure 1 survey. We added seven new taxa, including two seed plants (*Amborella*, *Thuja*), a fern (*Azolla*), three chlorophytes (*Raphidocelis*, *Chloropicon*, *Pycnococcus*) and a red alga (*Porphyridium*). These have no effect on our interpretations or conclusions.

Third, given that species you examined are highly diverse, using the blast search to identify homologs of system I and III genes could be problematic. You might first build hidden markov model on representative species (each can be selected for each clade) and then search the homologs of the system I and III genes.

Thanks for this comment, which further highlighted potential questions on this topic. As mentioned in our response to reviewer 1, we have made several modifications to address this perceived issue, which

had no effect on interpretation. Given the consistency of our survey results on various levels, we don't think a more advanced search is necessary.

2) In the main text, you mostly stated the system III. The system III can rescue yeast HCCS deletion lines, which is good. How about the systems I? Can systems I rescue yeast HCCS deletion lines? If not, it's problematic to come with a conclusion, which is that Archaeplastida lineages independently switched from system I to III

While this is an interesting idea, to rescue yeast HCCS deletion lines with system I would require the transgenic expression of perhaps up to 8 different system I genes. Also, many of these system I proteins (eg, *ccmB*, *ccmC*, and *ccmF*) are encoded in the mitochondrial genome, and probably for good reason as they have many transmembrane domains that would be unlikely to correctly orient if forced to be transgenically imported from the cytosol. Thus, we do not think this analysis has a high likelihood of success.

Regardless of the feasibility to test the ability of system I to complement a system III mutant, we think the more relevant question is whether a system III HCCS can complement a system I mutant. This is because the direction of evolution throughout eukaryotes is to switch from the ancestral system I to the eukaryote-specific system III. As described in detail in our response to reviewer 4, we have included our *Arabidopsis* complementation study (Figure 6), which revealed the successful rescue of an *Arabidopsis* system I mutant through the expression of fern HCCS.

3) The section "Phylogenetic analysis of eukaryotic HCCS homologs" might be unnecessary. If you really want to include it, you might need to reanalyze this. You could use the above approach (see concern #1) to screen system I and III genes in the genomes of species in major eukaryotic lineage (not all eukaryotic genomes), rather than in NCBI non-redundant protein database.

We appreciate these comments, which allow us to clarify our approach and reasoning. The purposes of the phylogenetic analyses are to evaluate the relationships among HCCS homologs, and to assess the level of possible HGT in its spread. While the NCBI non-redundant protein database may not have every eukaryotic genome sequence available, it certainly has a very good sampling of nuclear genomes. We were able to identify a large collection of >100 HCCS homologs from diverse eukaryotic lineages: alveolates, animals, archaeplastids, dictyostelids, fungi, haptophytes, and stramenopiles. These diverse HCCS homologs were analyzed in Fig 6A, showing (somewhat surprisingly to us) that the major lineages of eukaryotes seem to be monophyletic, suggesting that the HCCS gene was present early and transmitted vertically within these lineages. In Fig 6B, we reduced the data set to focus the analysis on major lineages of land plants and green algae, which we hoped would provide clarity for Archaeplastida on the two competing hypotheses about the origin and evolution of system III (ie, rampant HGT of system III HCCS, or ancestral presence of both systems followed by stochastic loss of one system). In essence, we feel that we have already performed the analysis of system III that the reviewer is suggesting.

With respect to system I, we feel that a phylogenetic analysis of the diversity of system I among diverse eukaryotes is less interesting, because the origin of system I is not really in question, as it must have

come from the alphaproteobacterial ancestor of the proto-mitochondrion. Phylogenetics is not needed to address this question.

Reviewer #3 (Remarks to the Author):

The manuscript reports the analysis on CCM system I and III. The authors comprehensively surveyed the presence/absence of CCM system I and II genes across diverse eukaryotes and found that sparsely distributed pattern of CCM system III correlates with the loss of CCM system I. They confirmed that putative system III homologs in viridiplantae indeed localize in mitochondria and is sufficient to complement in system III deficient yeast. Furthermore, using CRISPR-Cas9 system, they generated *Chlamydomonas* system III mutants and proved effects of system III on respiration. The authors performed phylogenetic analysis and proposed that both system I and III had existed in the archaeplastid common ancestor, but one of them were eventually lost, and several independent HGTs of system III within some land plant lineages followed after that. The manuscript is well written and the study is very interesting in that it reveals the distribution of CCM genes across the eukaryotes and discussed about the implications.

Thank you for summarizing our work and your valuable feedback.

However, fundamentally I am doubting the novelty of this study, as the critical results and discussions in the manuscript had already been reviewed in Allen, J.W.A., Jackson, A.P., Rigden, D.J., Willis, A.C., Ferguson, S.J. and Ginger, M.L. (2008), Order within a mosaic distribution of mitochondrial c-type cytochrome biogenesis systems?. *The FEBS Journal*, 275: 2385-2402. <https://doi.org/10.1111/j.1742-4658.2008.06380.x>. First, Allen et al. did a comparative genomics analysis and found that system I and system III are not compatible and system I genes are always bipartite into mitochondrial and nuclear genome. This statement is identical to the finding of the authors' study: mosaic distribution of CCM pathway.

We certainly agree that there is a good body of informative literature on this topic, including some of the early analyses presented in Allen et al 2008 (and others) and more recent studies (eg Babbitt et al 2015). And we have extensively cited the Allen et al 2008 study in our manuscript. However, we have shown that the loss of system I and switch to system III has occurred at least 11 times in Archaeplastida. None of these previous studies found the massively repeated evolution that has occurred in these lineages. Moreover, the Allen et al 2008 study primarily used mitogenomic data to infer the loss of system I, and in just a few species. This doesn't test for the possibility that these system I genes may have been transferred into the nucleus, an idea that we can clearly reject based on the results of our approach.

We also wish to point out that these many previous studies gave the impression that Charales (represented by one taxon in Allen et al 2008) and all land plants (represented by an unknown number of taxa in Allen et al 2008 as they are not listed in Allen's supp table 1) used system I, while all other green algae (represented by six mitogenomes) lost system I. However, we now know this is a huge oversimplification. Our much more comprehensive analysis includes not only mitogenomes, but also transcriptomes and some nuclear genomes to provide unambiguous evidence that the entire system I

pathway has been lost at least 11 times in Archaeplastida. Importantly, our vastly increased sampling of not just mitogenomic data but also transcriptomic and nuclear data demonstrates that these system I genes were truly lost and not transferred to the nucleus.

Finally, Allen et al 2008 also reports that Chlamy and a few other chlorophytes have a HCCS homolog. However, to be fair this wasn't an Allen 2008 discovery; the HCCS genes were already annotated in Chlamy by 2005. Importantly, our work alone has revealed substantial homologs to HCCS in many additional Archaeplastida lineages (not just chlorophytes), which have not been reported prior to our study. And most notably, our results show that there is a precisely parallel pattern of loss of system I and presence of system III in all 11 of these archaeplastid lineages. Thus, we feel that our study goes well beyond the previous results from Allen 2008 and other studies.

Second, Allen et al. performed a phylogenetic analysis of system III genes (heme lyases). Though with less taxa, different topology and the extent of lineages, development of the reasonings in both Allen et al and this manuscript resemble each other. Allen et al proposed two models: 1) origin in common ancestor of bikont and unikont, or 2) multiple LGT of system III gene (heme lyase), and concluded that model 2 is more likely to have occurred, unlike the author's statement that both systems presented in the Archaeplastida common ancestor (which resembles model 1).

Thank you for this comment. Because Allen et al 2008 (and Giege et al 2008) also proposed these models regarding the origin and spread of HCCS, we have attributed this idea by citing these two studies in the discussion (line 458). More generally, when any trait is sporadically present among taxa, there are essentially two competing hypotheses: the new trait may have been present in a common ancestor and repeatedly lost in lineages that do not have the trait today, or the new trait was gained by repeatedly gained to explain the sporadic presence among lineages. This same argument has been applied to many examples of repeated/convergent evolution: e.g., the multiple origins of C4 photosynthesis, the evolution of nodulation among legumes and related species, and the distribution of the mitochondrial *cox1* intron among flowering plants.

Third, Allen et al. discussed possible reasons that could explain the preference toward system III over system I. Here, the content is slightly different, but essentially, both discuss the same thing. Allen et al. proposed that no need of wide substrate specificity (as only need to mature two mitochondrial cytochromes c) gave selective force for system III. On the other hand, the authors said that complexity of the system I made simpler system III to thrive in most eukaryotes. However, those that still carry system I would benefit from enhanced regulatory control over the mitochondrion.

Thanks for this comment. We added a citation and a brief description of the Allen et al 2008 argument into the discussion, regarding the difference in number of targets in mitochondria vs. prokaryotes as a possible selective driver for the switch from system I to III (lines 525-529).

Allen et al. also mentioned similarity of between CCM system I&III and EF-1 α and EFL.

Thank you for reminding us that the Allen 2008 manuscript also mentioned the EF-1 α and EFL comparison. As there are still few comparable examples of enzyme/pathway switches that are not obviously adaptive, the EF-1 α and EFL references are definitely still relevant. Importantly, we already cited the EF-1 α and EFL studies in our manuscript.

The authors did a fine work to experimentally prove the function of putative system III genes in green algae. However, the results are not so much surprising because long have been known that the genes are system III genes. Though putative, the genes grouped well with experimentally proven system III genes in the phylogenetic tree and distinct substrate preferences of system III genes were already studied. In conclusion, the experiments the authors did were novel and appreciable, but these alone is not enough. The authors need to provide and clarify their novelty given their findings were already studied before.

Certainly, as we already acknowledge in the introduction, the Chlamy HCCS homologs were annotated as such in the nuclear genome, as early as 2005. However, homology alone is not a foolproof method for inferring function. Here, we performed the necessary experiments to validate the specialized function of the two algal HCCS homologs in mitochondrial respiratory chain and to characterize the general function of the single fern HCCS homolog. We feel the manuscript has already fairly assigned the novelty of previous work with respect to our work, but we are certainly open to discussing this issue further if there are other statements that may not be properly attributed.

In addition to addressing our evolutionary questions here, mitochondrial respiratory mutants have been long sought in green algal research. They are extremely important for studying organelle biology as well as its application in some emerging areas, such as the study of CO₂-concentrating mechanism (*Burlacot, A., Dao, O., Auroy, P. et al. Alternative photosynthesis pathways drive the algal CO₂-concentrating mechanism. Nature 605, 366–371 (2022)*). However, only a few such mutants have been obtained by general mutagenesis and most are not available/viable due to their defects on mitochondrial electron transport. Our CRISPR-generated mutants with validated functions in mitochondrial ETC complexes are valuable resources for the research community. For example, instead of using toxic inhibitory compounds to evaluate the respiratory complexes, our mutants hold value and advantages for mitochondrial experiments and biological systems without the potential side effects caused by adding chemical inhibitors. We have added a brief statement of the value of these validated mutants in the Results section (lines 303-306). In fact, we have already initiated collaborations to explore the potential uses of these novel mutants.

Other than that, I have few minor comments.

1. Line 42-43 “Convergent evolution describes the independent evolution of a similar trait in different species”: Add reference?

I think this is a fairly standard textbook definition. In this revised version, we cite a few studies that use essentially this same definition (line 40). As discussed in our response to other reviewer comments, we have opted to use the more general term ‘repeated evolution’, to avoid delving into the myriad definitions of convergent and parallel evolution.

2. Line 77-80: Provided references seem to cover only for plants, not eukaryotes.

Thanks for pointing out this discrepancy. We have modified the sentence (line 75) to refer to plants, as system I in eukaryotes is best studied in plants.

3. Line 80-84: is an interesting point for readers who are not familiar with ccm systems. Do a single system III gene perform all the roles of the eight system I genes? It would be helpful for readers if the functions of system III genes are described with the comparison to system I genes.

Thank you for this question. Based on the repeated evolutionary shifts from system I to III in eukaryotes, it is evident that the single HCCS enzyme of system III has the functional capacity to replace system I. Specifically, Babbitt et al 2015 summarized data supporting a 4-step model in which HCCS 1) binds heme, 2) binds apocytochrome *c*, 3) forms the thioether bond, and 4) releases mature holocytochrome *c*. These processes seem to align with the functions of CCMF and CCMH in system I. Whether HCCS performs similar roles of the other system I components is less clear. For example, in system I, heme transport is facilitated by CCMA, B, C to CCME. CCME, in turn, delivers heme to CCMF/H, ultimately leading to the ligation of heme to cytochrome *c*. However, this process remains less understood in system III. As you suggested, we have updated this sentence in the introduction to provide more information about HCCS function (lines 78-81).

As a side note, we are currently generating additional *Arabidopsis* mutants with disruptions to multiple system I genes to further clarify these relationships, but these efforts are time-consuming and would likely need to be presented in a future publication.

4. Line 156-157: Then who would be the donor?

The origin of HCCS is unclear. HCCS does not have a detectable prokaryotic homolog, suggesting it originated within eukaryotes (although it is possible this prokaryotic ancestry has eroded beyond detection). If HGT played a role in its spread among eukaryotes, then the donors of these transfers would be other eukaryotes. We have removed reference to “gain” at this point (lines 169-171), eliminating this point of uncertainty in the text.

5. Line 286-289: Please describe shortly why there are no differences under normal light conditions.

Chlamy is well known for its ability to grow mixotrophically, usually masking the respiratory defects of mitochondrial during photosynthetic growth. We have added a sentence (lines 298-301) with references to provide clarity on the absence of differences under normal light conditions.

6. Line 315: BS value 87 is not a low value but also, neither is strong. I would not say it is “strong”.

Thanks for this comment, which gets to the subjectivity in interpretation of bootstrap values. Although we feel a BS of 87% for a single gene is quite strong, we acknowledge that it is subjective. As discussed elsewhere, we have added Bayesian analyses and AU topology tests to provide statistically interpretable results for the branches and relationships in the tree.

7. Line 345-347: Proposed HGT is not convincing as single gene phylogeny often partially fail to correctly reconstruct phylogenetic relationships. The authors should remove statements regarding HGT within streptophytes, or provide more evidences.

Yes, it can be difficult to find unambiguously strong evidence for HGT in single-gene phylogenetics. Thus, we have added AU topology tests Bayesian posterior probabilities for additional evidence.

8. Line 459-462: There are many other lineages that carries system I. Will the same reasoning apply to the lineages other than seed plants? Or is it just simply because those lineages have never experienced HGT? Also, references that system I enhances mitochondrion control should be provided.

This is a speculative statement suggesting that increased complexity may be favorable by offering additional points for regulatory influence. We have updated this sentence (lines 518-524) and added references, which hypothesize that seed plants in particular may have additional layers of regulatory complexity to their mitochondria. As it is well known, seed plant mitochondria play an essential role during seed germination and early seedling development.

9. Figure 5: What does red arrows stand for?

The red arrows indicate HCS1 and HCS2 are involved in the maturation of cytochrome c and cytochrome c1 (subunit of the complex III), respectively. We added this to the Figure 5 description.

10. Figure 6 (also sup. fig. 3): It should be mentioned that BS value under 50 is not shown. Also, how is the tree rooted?

Thanks for catching this oversight. We added a statement in the figure 7 legend that BS < 50% and PP < 0.8 are not shown. The tree in Figure 7A was rooted using midpoint rooting. The tree in Figure 7B was rooted on red algae, in agreement with the larger eukaryotic analysis from Figure 7A.

11. Data availability: Alignments and treefiles must be provided.

Thanks for this comment. We have provided trimmed alignments and Newick tree files as a supplementary zip file.

12. Sup. Fig. 3: BS value 39 was not removed (see XP_005847167).

Thanks for catching this oversight. We have replaced this tree with an updated analysis, and we double-checked to ensure that all BS values <50% and Bayesian PP's <0.5 were removed. Thank you again for all comments.

Reviewer #4 (Remarks to the Author):

It remained unclear why and how mitochondrial CCM cytochrome maturation system I have been replaced by HCCS-mediated CCM cytochrome maturation system III multiple times independently in eukaryotic evolution. Li et al. tackled this issue by focusing on Archaeplastida. They re-investigated the phylogenetic distribution of system I and III in Archaeplastida and in eukaryotes. They found at least 11 independent evolutionary losses of system I genes in Archaeplastida. They also demonstrated that *Ceratopteris* and *Chlamydomonas* HCCSs could complement functions of the yeast homologue, indicating that the green HCCSs are indeed cytochrome synthases. On the basis of the implication of the phylogenetic analysis, the authors claimed that independent lateral gene transfers have shaped the current phylogenetic distribution of CCM and HCCS in Archaeplastida. The current work includes important clues to gain insight into functions and evolution of CCM and HCCS. However, all the biochemical, genetic, and phylogenetic analyses seem indirect or incomplete for the purpose. In addition, Discussion is comprised of large parts independent from data or evidence. I therefore suggest additional experiments and analyses.

We value this thoughtful evaluation from the reviewer and appreciate the suggested areas for improvement.

First, complementation of the yeast HCCS mutants by *Chlamydomonas* HCCS is the important step for clarifying HCCS functions and evolution in eukaryotes. However, the purpose of this study is to understand why and how CCM system I genes have been replaced by system III HCCS genes. If so, complementation of the CCM mutant of Archaeplastida (e.g., *Arabidopsis*) by a HCCS should be investigated. Whether CCM can be complemented by a HCCS and whether the complementation does not exhibit any growth impairment might provide insight into evolution of CCM/HCCS replacement.

Yes, we agree that this is an important issue. To explore this issue in detail, we had already begun experiments to investigate an artificial switch from system I to III in *Arabidopsis*. We introduced a fern system III HCCS into an *Arabidopsis* line with a heterozygous tDNA insertion in the nuclear *CcmH* gene (system I). After selfing this transgenic line, we identified homozygous *ccmH/ccmH* progeny (which were previously shown to be embryo-lethal (Meyer et al 2005)) that were rescued by expression of the fern system III HCCS. We now present this complementation study with phenotypic analysis of growth rate, seed germination and primary root growth. The results demonstrate that the homozygous system I mutant, when rescued with the fern system III HCCS, is indistinguishable from the wild type under optimal greenhouse conditions. Please refer to the new Figure 6 and the corresponding text in results (lines 319-366) for details on this experiment, and we added a point in the discussion (lines 513-514) about the implications of this result in understanding the evolution of this pathway shift.

Relevant to the above, it is not clear whether the authors have evaluated or decreased the possibility of off-target in the CRISPR-Cas9 experiment.

Regarding the specificity of CRISPR-Cas9 targeting, we used the Chopchop v3 (<https://chopchop.cbu.uib.no/>) for our gRNA design, which ranks potential target sites based on efficiency and off-target mismatches, by following the Chlamy gene editing protocol established (Akella et al 2021) and employed (Li et al 2023) in co-Author Cerutti's lab. In addition, the dark screening of CRISPR transformants for mitochondrial defects in this study further decreased the possibility of obtaining off-targeted mutants.

Akella S, Ma X, Bacova R, Harmer ZP, Kolackova M, Wen X, Wright DA, Spalding MH, Weeks DP, Cerutti H (2021) Co-targeting strategy for precise, scarless gene editing with CRISPR/Cas9 and donor ssODNs in *Chlamydomonas*. *Plant Physiol* 187:2637–2655

Li Y, Kim EJ, Voshall A, Moriyama EN, Cerutti H. 2023. Small RNAs >26 nt in length associate with AGO1 and are upregulated by nutrient deprivation in the alga *Chlamydomonas*. *Plant Cell* 35: 1868–1887.

Second, as mentioned by the authors, the phylogenetic tree is unresolved. Therefore, it is difficult to interpret anything from the tree.

While we agree that some parts of the trees are unresolved, we would disagree that it is difficult to “interpret anything from the tree”. We can certainly interpret some things, especially now with the Bayesian posterior probabilities provided throughout. First, we can see in Fig 7A that nearly all of the major eukaryotic groups are monophyletic with good support. This tells us that HCCS was present in the common ancestor of each of these groups. Second, we can see the clear separation of chlorophytes group 1 and group 2 in Figure 7A; while not every branch split has strong support, at least two branch splits do have good support (85/1.0 and 92/1.0) for this separation, so we can be confident that chlorophytes 1 and 2 are not derived from duplication. Finally, the Bayesian results and AU topology tests in the revised trees now provide more evidence for the phylogenetic incongruence within land plants, shown most clearly in Figure 7B.

Accordingly, the current Discussion is comprised of large part of assumption not derived from any data. For instance, lines 401-402 mention that “the two systems are essentially incompatible.” I could not find any evidence, references, or data for this. If the authors would like to suggest it, they should establish a transformant plant having both systems and investigate its growth and whether cytochrome maturation is impaired by the coexistence. The same transformant would also allow to investigate the authors' suggestions in lines 458-459, on benefit of system III.

Thanks for pointing out this unclear comment. Our comment that “the two systems are essentially incompatible” was not based on experimental data. It was an evolutionary inference from the fact that over evolutionary timescales, essentially all eukaryotes (with exception of two species in the newly described *Provora*) have only one or the other mitochondrial CCM pathway. But we agree that claiming the two systems are functionally incompatible is too strong. It is certainly possible that both pathways could coexist, but over evolutionary timescales one or the other pathway simply gets lost due to their

redundancy. We have revised this section (lines 461-468) to state that both systems are not maintained over long periods of evolutionary time (even in *Provora*, where only 2 of 7 described species have both systems).

In this point of view, we should not forget that any gene transfer events and thereby gene replacements always follow coexistence of an endogenous gene and an exogenous transferred gene for a certain evolutionary period, prior to loss of the endogenous one. This is because the endogenous gene cannot be non-functionalized or lost prior to functionalization of the transferred gene for the cell viability. Thus, both systems should have functioned in a same cell although it is difficult to imagine and evaluate how long they have coexisted. Gene transfer events of HCCS might imply that coexistence of systems I and III are not incompatible for a certain term.

Yes, we fully agree with this comment. For the switch to occur successfully, both systems must coexist for some period of time. However, the two hypotheses make very different inferences about the length of time that both must coexist. In the HGT scenario, both systems might coexist for as little as a few generations, whereas for the ancestral presence and differential loss scenario, both systems would need to coexist in some lineages for hundreds of millions of years. As mentioned in the previous response, we have revised this section to talk about the evolutionary evidence that these two systems are not maintained over evolutionary timescales, rather than giving the impression of their absolute incompatibility.

Further, in line 406-420, since the tree is not resolved well, the tree topology does not provide any clue to gain insight into evolution of HCCS. So are lines 429-435. For the discussion, the authors could statistically evaluate whether the alternative tree showing each of ferns, lycophytes, and hornworts is monophyletic is rejected or not by the AU test. If rejected, the authors' scenario might in part be supported.

Thanks very much for this suggestion. As discussed above, although the tree is not fully resolved, we can nevertheless make some claims confidently, including the monophyly of most eukaryotic groups, and the separation of chlorophytes group 1 and 2. Specifically with regard to the statements about HGT in ferns and lycophytes, as discussed elsewhere, we have provided additional analyses, including Bayesian phylogenetics and AU topology tests, to bolster our claims of possible HGT.

lines 156-157

I could not follow this. The current distribution can also be explained by differential losses after coexistence of system I and III by which “extant species” have completely lost either of them. I do not say the authors' claim and the evolutionary scenario are wrong. But I would say the authors' claim is not based on any data and therefore which evolutionary scenario is more likely cannot be evaluated quantitatively. Currently, both scenarios are not rejected.

We agree that there are two hypotheses, and neither can be rejected. However, the two hypotheses should differ with respect to phylogenetic results. Under the differential loss hypothesis, phylogenetic

analysis of the HCCS genes should follow organismal patterns for the species that have system III. Under the HGT hypothesis, these HCCS genes would not follow organismal patterns. More indirectly, the differential loss hypothesis requires that both systems I and III would need to be present for much longer periods of time than the HGT hypothesis. The fact that only one eukaryotic lineage (Provora) out of hundreds examined seem to have both systems, suggests that both systems are not maintainable for long periods of time.

As mentioned above, we have added statistical support from Bayesian analysis and AU topology tests for the lack of monophyly of ferns and lycophytes, which provides additional data for the HGT hypothesis.

Fig1 and Fig6 are inconsistent in light of taxon sampling. Some species included in Fig6 are not in fig.1 such as *Dicranopteris*. Since Fig.1 is the source data for Fig. 6, they should be consistent to each other.

Yes, there is a discrepancy here. The more limited taxon sampling in Fig 1 relative to Fig 6 stems from the previously discussed fact that Fig 1 sampling required a mitogenome sequence. Thus, we could assemble a nuclear transcriptome for some species (including many ferns), enabling us to get HCCS sequences for Figure 6, but we could not include them in figure 1 because they have no available mitogenome. All of these new HCCS sequences were deposited in GenBank (Accessions BK063718–BK063750). Indeed, while this may not have been obvious from the manuscript, we assembled draft mitogenomes [using either available SRA data or our own new sequencing data (SRA accessions SRR24748178–SRR24748183)] for many ferns and several lycophytes included in fig 1, which provided the needed mitogenomic data for inclusion in Fig 1.

For some groups (eg, flowering plants and chlorophytes) we chose to limit the number of taxa shown in figure 1, which was done simply to maintain a compact size for figure 1. We could certainly add many more angiosperms and chlorophytes to Fig 1, but they would also not add any useful information that changes the interpretations. For example, we decided to revise figure 1 to more taxa, including two seed plants (*Amborella*, *Thuja*), three chlorophytes (*Raphidocelis*, *Chloropicon*, *Pycnococcus*), and a red alga (*Porphyridium*), but these additions have absolutely no effect on interpretation of the frequency or timing of system I to III switches.

Reviewers' Comments:

Reviewer #1:

Remarks to the Author:

The authors have addressed the many interrelated reviewer comments and I believe the manuscript to be strengthened as a result. The evolutionary patterns are remarkable and I hope the report spurs additional study.

Reviewer #2:

Remarks to the Author:

I appreciate the authors for addressing my concerns. I no longer have any concerns and recommend the acceptance of the publication in Nature Communications.

Reviewer #3:

Remarks to the Author:

The authors adequately addressed all the comments I raised. I believe this manuscript is now worthy of publication.

Reviewer #4:

Remarks to the Author:

All the concerns raised for the submitted manuscript have been well addressed by additional analyses and experiments in the revised manuscript. I have no additional comment to the manuscript.